# MIND OVER BODY: ADAPTIVE THINKING USING DYNAMIC COMPUTATION

**Mrinal Mathur**[*]
TReNDS Center[†]
Georgia State University
Atlanta, GA, USA

**Barak A. Pearlmutter**[*]
Dept of Computer Science
Maynooth University
Co. Kildare, W23 A3HY, Ireland

**Sergey Plis**[*]
TReNDS Center[†]
Georgia State University
Atlanta, GA, USA

## ABSTRACT

While the human brain efficiently handles various computations with a limited number of neurons, traditional deep learning networks require a significant increase in parameters to improve performance. Yet, these parameters are used inefficiently as the networks employ the same amount of computation for inputs of the same size, regardless of the input's complexity. We address this inefficiency by introducing self-introspection capabilities to the network, enabling it to adjust the number of used parameters based on the internal representation of the task and adapt the computation time based on the task complexity. This enables the network to adaptively reuse parameters across tasks, dynamically adjusting computational effort to match input complexity. We demonstrate the effectiveness of this method on language modeling and computer vision tasks. Notably, our model achieves 96.62% accuracy on ImageNet with just a three-layer network, surpassing much larger ResNet-50 and EfficientNet. When applied to a transformer architecture, the approach achieves 95.8%/88.7% F1 scores on the SQuAD v1.1/v2.0 datasets at negligible parameter cost. These results showcase the potential for dynamic and reflective computation, contributing to the creation of intelligent systems that efficiently manage resources based on input data complexity.

## 1 INTRODUCTION

The complexity and scale of deep learning models have skyrocketed, propelling significant advancements across diverse fields involving images, text, and even robots. However, adaptation of computation to problem difficulty is still one of the most challenging aspects in deep learning. Traditional architectures process inputs through a fixed number of layers, which can waste computational resources on simple tasks or be insufficient for more complex ones (Canziani et al., 2016; Wang et al., 2016; Bai et al., 2023). This one-size-fits-all approach does not account for the varying difficulties of input data, leading to inefficiencies and suboptimal performance (Huang et al., 2017b; Fregoso-Aparicio et al., 2021).

Drawing inspiration from the human brain's dynamic allocation and reuse of neurons to efficiently handle multiple tasks (Schulz & Gershman, 2019), we propose the Model INtrospection for a Dynamically adaptive (MIND) model. The MIND model includes two networks: the primary prediction network and an auxiliary introspection network. By assessing the representation of each input in the prediction model, the introspection network determines the computational capacity to employ. This process involves selecting the number of layers to iterate through in the prediction model, thereby adapting the computational effort to the complexity of the input.

---

[*]Correspondence to: mmathur4@gsu.edu, barak@pearlmutter.net, s.m.plis@gmail.com

[†]The Georgia State University/Georgia Institute of Technology/Emory University Center for Translational Research in Neuroimaging and Data Science (TReNDS Center).

Unlike conventional models, which scale parameters indiscriminately (Jeon et al., 2021; Lu et al., 2022; Wu et al., 2020), our model adaptively reuses parameters across tasks. By doing so, it balances the inherent tension between the efficient use of parameters across tasks of varied complexity and the challenge of parameter allocation based on input difficulty.

The introspection network operates as a switch, choosing between the Fixed-Point Iterative (FPI) layers or opting for a no-operation action, based on the input representation in the prediction model (Vaswani et al., 2017; Hochreiter & Schmidhuber, 1995). Notably, given that the introspection network's decision is based on the input representation produced by the more complex prediction network, it does not need as much complexity as that required for a direct input assessment.

In essence, the MIND model effectively manages the allocation of computational efforts, saving complex tasks for more comprehensive processing and conserving resources for simpler ones. Figure 1 outlines the architecture of our proposed model.

To demonstrate the effectiveness of this approach, we show how the MIND model can detect patterns of varying difficulty using an Ising model (Cipra, 2000) as a toy example, illustrating its adaptive computational capabilities. We further validate the MIND model's effectiveness on standard benchmarks across multiple domains. In language modeling, we evaluate on SQuAD (Rajpurkar et al., 2016) and WikiText (Gardent et al., 2017). To demonstrate that it is domain-agnostic, we also validate on vision benchmarks including CIFAR-100 (Krizhevsky, 2009) and ImageNet (Deng et al., 2009). Remarkably, the MIND model surpasses ResNet-50 (He et al., 2016) and EfficientNet (Tan & Le, 2021) on ImageNet, and Transformer (Vaswani et al., 2017) on SQuAD, while using only a three-layer predictive network and 5 to 12 times fewer parameters.

The remainder of this paper is structured as follows: Section 2 provides an overview of background work related to adaptive computation and

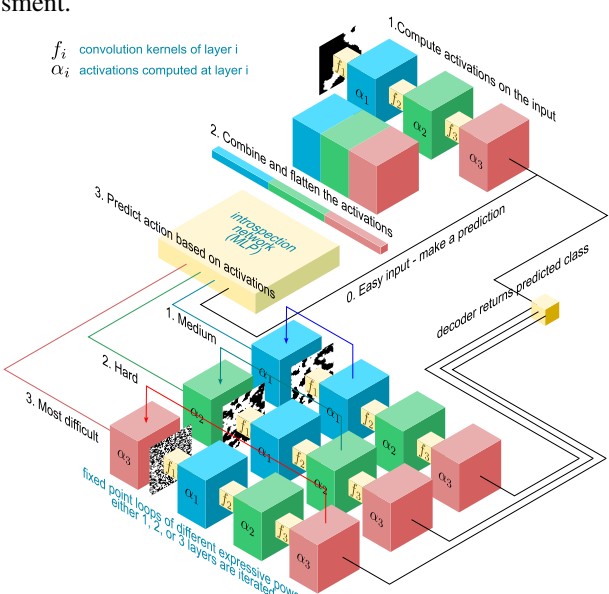

Figure 1: Architecture of the MIND model and the order of operation. The introspection network decides on the computational branch to follow, optimizing efficiency by allocating more resources to harder inputs and less to easier ones. FPIs reuse parameters and dynamically adapts to the input difficulty.

introspection mechanisms in deep learning; Section 3 details the architecture of the MIND model, explaining the roles of the primary prediction network and the auxiliary introspection network; Section 4 presents our experimental methodology, the datasets used, and the results obtained from our evaluations.

## 2 RELATED WORK

### 2.1 FIXED-POINT METHODS IN NEURAL NETWORKS

Fixed-point iteration (FPI) methods have a rich history in neural network optimization and architecture design. Both Almeida (1987) and Pineda (1987) independently introduced the concept of fixed points by extending backpropagation to recurrent neural networks. Recent advances, such as the modernized formulation of Liao et al. (2018) and the Deep Equilibrium Models (DEQs) proposed by Bai et al. (2019), build on these early insights. DEQs implicitly define infinitely deep networks through a fixed-point formulation, continuing the tradition of using FPI methods in neural networks. Our approach integrates these developments, incorporating FPI layers to enable adaptive computation depth.

FPI are designed to find a fixed point $z$ that satisfies $z = f(z, x)$. Here, $f$ is the neural network layer's function, $x$ is the input to the layer, and $z$ is the fixed-point we compute by iteration, $z^{(k+1)} = f(z^{(k)}, x)$, where $z^{(k)}$ is the estimate at iteration $k$. The process begins with an initial guess $z^{(0)}$ and iterates until convergence, $\|z^{(k+1)} - z^{(k)}\| < \epsilon$ where $\epsilon$ is a small tolerance. Furthermore, Liao et al. (2018) shows a crucial theorem:

$$z = J_F^{\top, h^*} z + \left( \frac{\partial L}{\partial y} \frac{\partial y}{\partial h^*} \right)^{\top}. \tag{1}$$

The equation determines the new state $z$ by applying a transformation to the current state, adjusted by transposed gradients of the loss with respect to outputs and parameters reduces memory usage during backpropagation by not storing all intermediate vectors. The theoretical underpinnings of FPI in neural networks have been further solidified by works such as Barrett & Bolt (2024), who developed frameworks for differentiating through nonsmooth iterative algorithms.

### 2.2 ADAPTIVE COMPUTATION AND INTROSPECTIVE NEURAL ARCHITECTURES

The idea of dynamically adjusting computational effort based on input complexity has gained significant traction in recent years. Granas & Dugundji (2003) introduced Adaptive Computation Time (ACT) for recurrent neural networks, enabling models to learn the optimal number of computational steps. Building on this, Figurnov et al. (2017) extended the concept to spatially adaptive computation time for image recognition tasks. Subsequent approaches have explored various mechanisms for adaptive computation. Huang et al. (2016) proposed stochastic depth networks that randomly drop layers during training, implicitly creating an ensemble of networks with varying depths. Similarly, Huang et al. (2017a) introduced dense networks with early-exit branches, allowing for dynamic inference paths. Banino et al. (2021) developed PonderNet, which learns to adapt the number of computational steps dynamically, while Elbayad et al. (2019) introduced depth-adaptive transformers for efficient language processing. Recent advances in dynamic layer selection, such as DynaLay, which introduces an introspective approach for selecting layers in deep networks Mathur & Plis (2023) and Mixture of Depth (MOD) Raposo et al. (2024) dynamically adjusts processing depth based on input complexity, optimizing computation in CNNs and Transformers. While MoD selects key channels or token routes, MIND model extends this concept by employing fixed-point iteration (FPI) and introspection to dynamically adapt both layer depth and computation. Unlike MoD's static depth adjustments, MIND model's iterative refinement enables more granular control, enhancing efficiency and performance across varying input complexities.

Our MIND model builds upon these ideas by incorporating a dedicated introspection network that analyzes activation patterns to make informed decisions about computational paths. This approach conceptually relates to the meta-learning framework proposed by Andrychowicz et al. (2016), where a separate network learns to optimize the prediction network.

## 3 MIND MODEL: DEEP LEARNING MODEL WITH INTROSPECTION

We present the MIND model framework, illustrated in Figure 1, which introduces an introspective mechanism in a combination with FPIs into deep learning models to achieve adaptive computation. The MIND model consists of two main components: the **introspection network** and the **prediction network**. The introspection network serves as the central unit that assesses the complexity of the input and the current activation states, dynamically adjusting the computational graph of the prediction network to optimize resource allocation and computational efficiency for specified tasks.

### 3.1 INTROSPECTION NETWORK ARCHITECTURE

The introspection network $\mathcal{I}$ is the core contribution of our framework, responsible for analyzing intermediate activations and determining the computational pathway within the prediction network $\mathcal{P}$. By dynamically adjusting the computational graph based on input complexity, the introspection network enables the MIND

model to allocate computational resources efficiently, dedicating more effort to complex inputs and less to simpler ones.

Given an input $x \in \mathcal{X} \subseteq \mathbb{R}^{d_x}$ and the activations $z_{l\,l=1}^{N}$ from the layers $\mathcal{L} = L_1, L_2, \ldots, L_N$ of the prediction network, the introspection network maps these activations to a layer selection mask $\mathbf{m} = [m_1, m_2, \ldots, m_N] \in {0, 1}^N$. Here, $m_l = 1$ indicates that layer $L_l$ requires more computation via fixed-point iterations, while $m_l = 0$ means standard forward propagation is sufficient.

It aggregates activations from selected layers $\mathcal{A} \subseteq 1, 2, \ldots, N$ of the prediction network to form a feature representation: $a = \phi\left(\{z_l\}_{l \in \mathcal{A}}\right)$, where $\phi$ is a function that processes and concatenates the activations, such as global average pooling followed by flattening. Using this feature representation $a$, the introspection network computes a decision vector: $\mathbf{d} = \mathcal{I}(a; \theta_{\mathcal{I}})$, where $\theta_{\mathcal{I}}$ are the learnable parameters of the introspection network. The decision vector $\mathbf{d}$ represents a probability distribution over possible layer selections. The final layer selection mask $\mathbf{m}$ is obtained by sampling or taking the argmax over the probabilities: $\mathbf{m} = \arg\max_{\mathbf{m}'} p(\mathbf{m}' \mid a)$

To allow for stochasticity and exploration during training, we used the Rao-Blackwell (Blackwell, 1947) straight-through Gumbel-Softmax (Paulus et al., 2020) estimator to determine the probability distribution that gives correct representation of states.

## 3.2 FIXED-POINT ITERATION MECHANISM.

Upon receiving an input $x_i$, the introspection network $\mathcal{I}$ estimates the complexity of the input by analyzing the activations of the prediction network. Based on this assessment, it determines the subset of layers that should be engaged and applies fixed-point iterations to these layers. This process allows the model to iteratively refine its computations, dedicating more resources to complex inputs that require deeper processing and fewer to simpler inputs. The iterative update rule for the fixed-point iteration is $x^{(l+1)} = f\left(x^{(l)}, \theta_l\right)$, where $x^{(l)}$ is the input to layer $l$, $\theta_l$ represents the learnable parameters of that layer, and $f$ denotes the non-linear activation function. The iteration continues until convergence is achieved, determined by the convergence criterion $\|x^{(l+1)} - x^{(l)}\| / \|x^{(l+1)}\| < \epsilon$ or until a maximum number of iterations $K_{\max}$ is reached. We have shared complete proof in Appendix B. Here, $\epsilon$ is a predefined tolerance level that controls the precision of convergence. Algorithm 3 outlines the forward propagation process with fixed-point iterations in the prediction network. By incorporating fixed-point iterations, the prediction network can adaptively allocate computational resources. This mechanism enhances computational efficiency without compromising performance, enabling the MIND model to handle a wide range of input complexities effectively. The ability to dynamically adjust the depth of computation allows the model to process simple inputs quickly while dedicating more computational effort to complex inputs that require it.

## 3.3 PREDICTION NETWORK ARCHITECTURE

The prediction network is designed to integrate seamlessly with standard deep learning architectures, supporting fixed-point iterative computations as guided by the introspection network. It processes the input data to generate predictions, and its internal activations are analyzed by the introspection network to dynamically adjust computational efforts based on the complexity of each input. We have shared how activation maps are calculated in Appendix D.

We show two agnostic features: one for vision tasks using a Convolutional Neural Network (CNN), and another for language tasks using LSTM and Transformer architectures. The MIND model enables adaptive computation akin to human cognitive processes, where the brain allocates resources efficiently depending on task difficulty (Marblestone et al., 2016). Traditional deep learning models apply a fixed computational graph to all inputs, leading to inefficiencies when processing data with varying complexities (Graves, 2016).

The prediction network is designed for seamless integration with standard deep learning architectures, supporting fixed-point iterative computations to handle inputs of varying complexity without increasing the parameter count.

**Vision Tasks.**  For computer vision applications, we implement the prediction network using a lightweight convolutional neural network (CNN) architecture. Specifically, it consists of three convolutional layers with filter sizes of 64, 128, and 256 channels, respectively. Each layer employs a kernel size of $3 \times 3$ and is followed by a Rectified Linear Unit (ReLU) activation function (Xu et al., 2015) to introduce non-linearity. Batch normalization is applied after each activation to stabilize the learning process (Ioffe & Szegedy, 2015). This architecture strikes a balance between computational efficiency and representational capacity, containing approximately 5.6 million parameters, which facilitates rapid training and inference while handling complex image data effectively.

**Language Tasks.**  In the context of natural language processing, the prediction network is instantiated as a dual-layer Long Short-Term Memory (LSTM) network (Hochreiter & Schmidhuber, 1995). Each LSTM layer consists of 256 hidden units, capturing temporal dependencies in sequential data. Dropout layers are interleaved between the LSTM layers to prevent overfitting, enhancing the model's generalization capabilities (Srivastava et al., 2014).

**MIND-Transformer**  For natural language processing tasks, we extend the prediction network to a Transformer architecture, introducing adaptive computation in both the self-attention and feed-forward networks. The self-attention mechanism with fixed-point iterations $f_\theta$ is defined as:

$$A_0 = \text{softmax}\left(\frac{QK^T}{\sqrt{d}}\right) V$$

$$A_{k+1} = f(A_k, x; \theta) = \text{softmax}\left(\frac{QK^T + f_\theta(A_k)}{\sqrt{d}}\right) V \quad (2)$$

where $Q = W_Q x$, $K = W_K x$, $V = W_V x$ are the query, key, and value projections of the input $x$, and $f_\theta$ is a learnable function that refines the attention mechanism.

Table 1: MIND-Transformer Details

| Component | Setting |
|---|---|
| Number of Layers | 6 |
| Dimension ($d_{\text{model}}$) | 512 |
| Attention Heads # | 8 |
| FFN Dimension | 2048 |
| Max Sequence Length | 512 |
| Fixed Point Iteration | 1-6 layers |

Similarly, we apply fixed-point iterations $g_\theta$ in the feed-forward network:

$$\text{FFN}_0(x) = W_2 \cdot \text{ReLU}(W_1 x + b_1) + b_2$$
$$\text{FFN}_{k+1}(x) = W_2 \cdot \text{ReLU}(W_1 x + b_1 + g_\theta(\text{FFN}_k(x))) + b_2$$

Furthermore, we cap the number of iterations in all FPIs based on the input complexity score computed as:

$$\text{IC}(x) = \alpha \cdot (1 - \max(\text{softmax}(f(x)))) + \beta \cdot H(\text{softmax}(x)) + \gamma \cdot \|\nabla_x f(x)\|_2, \quad (3)$$

where $f(x)$ is the model's output before the final softmax layer, $H(.)$ is the entropy function, $|\nabla_x f(x)|_2$ is the $L_2$ norm of the input gradient and $\alpha$, $\beta$, and $\gamma$ are weighting coefficients set to 0.4, 0.4, and 0.2 respectively. Maximum number of iterations is set to $\max(10 \, \text{IC}(x), 50)$.

For simplicity, the MIND-Transformer employs the same configurations as a standard Transformer of Vaswani et al. (2017) (see Table 1). The model incorporates fixed point iterations within its self-attention mechanism and transition function block ($\phi$) across multiple layers. We utilize relative positional embedding with a sequence length of 120 tokens for both training and inference processes.

### 3.4  TRAINING THE MIND MODEL

Training the MIND model involves jointly optimizing the prediction network and the introspection network to achieve high predictive performance while efficiently allocating computational resources. This section details the training objectives, loss functions, optimization strategies, and the methodology for computing gradients through the fixed-point iterations using phantom gradients as given in Algorithm 1 In Appendix A. The overall training objective is to minimize a composite loss function that balances prediction accuracy

against computational efficiency:

$$\mathcal{L}_{\text{total}} = \mathcal{L}_{\text{pred}} + \lambda \mathcal{L}_{\text{introspect}}, \tag{4}$$

where, $\mathcal{L}_{\text{pred}}$ is the prediction loss from the prediction network, $\mathcal{L}_{\text{introspect}}$ is the introspection loss from the introspection network, penalizing computational cost, $\lambda$ is a hyperparameter balancing the influence of the introspection loss. The prediction loss $\mathcal{L}_{\text{pred}}$ measures the discrepancy between the model's predictions and the ground truth labels (e.g. cross entropy). The introspection loss $\mathcal{L}_{\text{introspect}}$ encourages the efficient use of computational resources by penalizing the use of additional layers and iterations,

$$\mathcal{L}_{\text{introspect}} = \frac{1}{M} \sum_{i=1}^{M} \left( \beta \cdot \text{CompCost}_i + \gamma \cdot \sum_{l=1}^{N} w_l \cdot m_{i,l} + \delta \cdot \sum_{l=1}^{N} m_{i,l} \right), \tag{5}$$

where $\text{CompCost}_i$ is the computational cost for input $x_i$, calculated based on the number of layers and iterations used, $m_{i,l} \in \{0, 1\}$ indicates whether layer $l$ is selected for input $x_i$, $w_l = \frac{c_l}{\sum c_l}$ is the importance weight for layer $l$, reflecting its computational cost, $\beta$, $\gamma$, and $\delta$ are hyperparameters controlling the trade-off between accuracy and efficiency. Ablation of these hyperparameters in explored in Appendix F.3.

The layer selection variables $m_{i,l}$ are discrete, introducing non-differentiability into the optimization process. To enable gradient-based optimization, we employ the Gumbel-Softmax trick (Jang et al., 2017; Paulus et al., 2020) to approximate the discrete sampling with a differentiable operation. For each input $x_i$, the introspection network outputs logits $\mathbf{z}_i = [z_{i,1}, \ldots, z_{i,N}]$, from which we compute the selection probabilities:

$$p_{i,l} = \frac{\exp\left((z_{i,l} + g_{i,l})/\tau\right)}{\sum_{j=1}^{N} \exp\left((z_{i,j} + g_{i,j})/\tau\right)}, \tag{6}$$

where $g_{i,l}$ are samples from the Gumbel(0,1) distribution, and $\tau$ is the temperature parameter controlling the smoothness of the approximation. This continuous relaxation of $m_{i,l}$ is then used in place of the discrete variables during optimization. The prediction network applies fixed-point iterations in selected layers to refine the activations. Backpropagating through these iterations poses challenges due to their implicit nature.

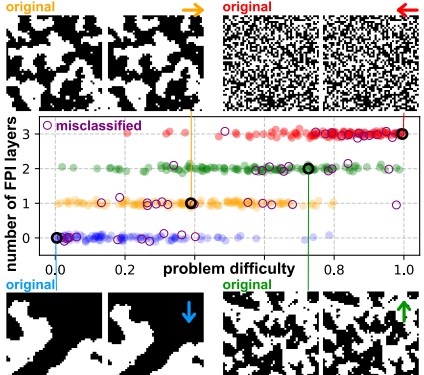

Figure 2: MIND model performance on the random dot movement across various consolidation difficulty in test data. Harder problems are adaptively addressed with more layers in the FPI.

## 4 EXPERIMENTS

We evaluated MIND against existing deep learning models using Vision and NLP benchmarks covering more than 10 datasets. For robustness, all experiments are conducted using 10 different random seeds and 9-fold cross-validation. Additional training specifications and extensive details on the datasets are given in Appendix C. We want to provide a rich picture of how MIND performs in different scenarios. The primary goals of these experiments are twofold: 1) to evaluate the classification accuracy of our model in comparison with existing state-of-the-art architectures, and 2) to assess the computational efficiency improvements achieved through our adaptive layer selection mechanism. We also share additional experiments in Appendix E including Ablations.

### 4.1 TOY EXAMPLE: LEVELS OF PROBLEM COMPLEXITY

To evaluate and demonstrate how the MIND model handles inputs of varying complexities, we used a problem where the degree of difficulty is apparent to a human and also closely aligns with the performance of a deep learning network. This example was inspired by the motion coherence task, also known as the random dot motion task (Newsome et al., 1989).

In this task, an animal is presented with a display of moving dots, some of which move randomly while some move in a coherent direction. The animal must identify the direction of the coherent motion. This task has been shown to elicit consistent results across different animals, including pigeons (Bischof et al., 1999), rats (Reinagel, 2013), monkeys (Newsome et al., 1989), and humans (Heekeren et al., 2004). To create a meaningful evaluation within the realm of predictive models and convolutional neural networks (CNNs), we adapted this task into a two-image input scenario. The CNN would receive an original image and its shifted counterpart — the shift can be to the left, right, up, or down.

For varying levels of difficulty, we used dot consolidation, a widely-utilized approach, and elected to present states of an Ising model at different temperatures. The difficulty of these levels ranges from a fully random state (hardest) to a highly consolidated, low-temperature state (easiest). Figure 2 provides four sample scenarios with varying degrees of difficulty. Notably, the number of layers in the FPI is clearly correlated with the problem's complexity, and errors are randomly distributed across the difficulty levels. This demonstrates how the MIND model successfully adjusts computation based on input complexity. The MIND model was able to reach an accuracy of $0.85 \pm 0.007$, while a CNN with the same number of layers and channels as the prediction network could only achieve an accuracy of $0.56 \pm 0.0004$ on this 4-class task with variance across a 9-fold cross-validation.

## 4.2 LAYER UTILIZATION ANALYSIS

To evaluate the introspection mechanism of the MIND model, we analyzed how the model allocates computational resources based on input complexity. We categorized the ImageNet dataset into three complexity levels—*easy*, *medium*, and *hard*—using the confidence scores from a pre-trained ResNet-50 model (He et al., 2016). Specifically, inputs with softmax probabilities above 0.8 were labeled as *easy*, those between 0.4 and 0.8 as *medium*, and those below 0.4 as *hard*.

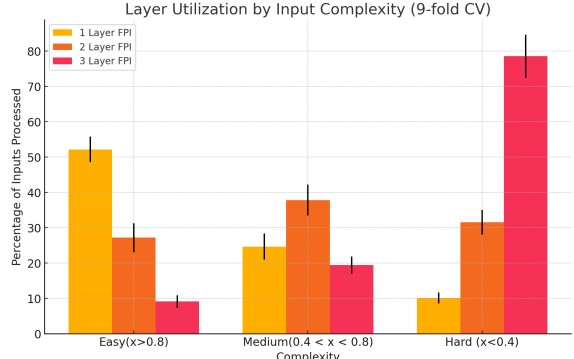

Figure 3: Analysis of layer utilization for samples from the ImageNet dataset, categorized into three complexity levels: easy, medium, and hard. The figure illustrates how the MIND model allocates computational resources based on input complexity, with more layers being utilized for harder examples.

The frequency distribution of the number of layers used across different input complexity levels as illustrated in Figure 3 demonstrates that the MIND model adapts its computational effort according to input complexity. Inputs classified as *easy* predominantly trigger minimal computation, often utilizing only the first layer in a convergent loop. In contrast, more complex inputs engage additional layers, indicating the model's capacity to dynamically allocate resources when faced with challenging tasks. This adaptive behavior validates the effectiveness of the introspection mechanism in assessing input complexity and adjusting computation accordingly. Due to limited space, relevant experiments are also shown in Appendix E.1.

The MIND model achieves superior performance on both datasets while using significantly fewer parameters compared to the baseline models. Notably, on ImageNet, the MIND model attains a Top-1 accuracy of 88%, outperforming EfficientNet-B7 by 3.7 percentage points despite having approximately 12 times fewer parameters. Additionally, the MIND model demonstrates enhanced robustness, which we attribute to its adaptive computation and ability to allocate resources effectively based on input complexity.

## 4.3 EXPERIMENTS ON VISION TASKS

To evaluate the effectiveness of the MIND model in vision tasks, we implemented a 3-layer Convolution Neural Network (CNN) (LeCun et al., 2015) as the prediction network within the MIND framework. We compared our model against traditional baselines and state-of-the-art architectures, specifically ResNet-50 (He et al., 2016) and EfficientNet-B7 (Tan & Le, 2021), which are renowned for their high performance

Table 2: Performance comparison of the MIND model with ResNet-50 and EfficientNet-B7 on CIFAR-100 and ImageNet. The MIND model parameter count is shown as "Prediction network/Introspection network/Total".

| Model | #Parameters | Dataset | Top-1 Accuracy ↑ | Top-5 Accuracy ↑ | Multi-label Accuracy ↑ | Robustness ↑ |
|---|---|---|---|---|---|---|
| ResNet-50 | 25.6M | CIFAR-100 | 69.9% | 84.7% | — | — |
| | | ImageNet | 74.3% | 91.6% | 68% | 69% |
| EfficientNet-B7 | 66M | CIFAR-100 | 84.4% | 91.1% | — | — |
| | | ImageNet | 84.3% | 95.56% | 75% | 81% |
| **MIND model** | 5.01M/0.3M/**5.31M** | CIFAR-100 | **85.53%** | **92.6%** | — | — |
| | | ImageNet | **88.3%** | **96.62%** | **78%** | **83%** |

in image classification tasks. We evaluated our models on CIFAR-100 (Krizhevsky, 2009) and ImageNet (Deng et al., 2009) datasets models using Top-1 and Top-5 accuracy metrics. For ImageNet, we also assessed multi-label accuracy (Yun et al., 2021) and robustness to ImageNetV2 (Recht et al., 2019). Table 2 shows that the MIND model achieves the highest Top-1 and Top-5 scores on both datasets. Note that it outperformed both ResNet and EfficientNet-7 on multi-label accuracy and robustness metrics on ImageNet. These results demonstrate superior performance of the MIND model across different datasets and metrics, highlighting its effectiveness in vision tasks. We also show MIND's performance on different vision datasets in Appendix E.5.

Table 3: Performance comparison of different models on WikiText and SQuAD v2.0 datasets. Perplexity (PPL) and bits-per-character (BPC) are reported for language modeling on WikiText-2 and WikiText-103. F1/EM scores are used for SQuAD v2.0.

| Model | WikiText-2↓ | | WikiText-103 (PPL)↓ | SQuAD v2.0 (F1/EM)↑ | Params (M) ↓ |
|---|---|---|---|---|---|
| | PPL | BPC | | | |
| LSTM (Yu et al., 2019) | 99.3 | — | 48.7 | — | 20 |
| LSTM-MIND (ours) | **14.8** | **0.85** | **30.5** | **72.7 / 70.5** | **8.6** |
| Transformer (Vaswani et al., 2017) | 29.2 | 1.04 | 18.3 | — | 110 |
| BERT-base (Devlin, 2018) | — | — | — | 76.8 / 73.6 | 110 |
| RoBERTa-base (Liu, 2019) | — | — | — | 83.7 / 80.5 | 125 |
| MIND-Transformer (Ours) | **14.5** | **0.80** | **16.3** | **88.7/ 81.01** | **112** |

## 4.4 EXPERIMENTS ON LANGUAGE MODELING TASKS

We evaluated performance on language modeling tasks across multiple datasets including WikiText-2 and WikiText-103 datasets (Merity et al., 2016). We compared the MIND model with baseline LSTM models (Yu et al., 2019) and state-of-the-art Transformer-based models (Vaswani et al., 2017; Dai et al., 2019). Table 3 presents the perplexity (PPL) and bits-per-character (BPC) results for WikiText-2 and WikiText-103 datasets, along with the F1 and Exact Match (EM) scores for SQuAD v2.0. The LSTM-MIND model significantly reduces perplexity on WikiText-2 (PPL: 14.8) and achieves competitive results on WikiText-103 (PPL: 30.5), outperforming the standard LSTM baseline. Similarly, the MIND-Transformer achieves superior performance across all tasks, with a notable improvement in SQuAD v2.0 (F1: 88.7%, EM: 81.01%) compared to both BERT-base (F1: 76.8%, EM: 73.6%) and RoBERTa-base (F1: 83.7%, EM: 80.5%). Table 7 shows the performance of the LSTM-based MIND model on SQuAD1.1, where it achieves a 95.4% F1 score with confidence interval of 0.2%, further highlighting its strong performance across different datasets.

The MIND-Transformer's results demonstrate its ability to outperform leading transformer models in both perplexity and downstream question-answering tasks, despite utilizing fewer parameters (112M compared to RoBERTa's 125M). These results highlight the efficiency of the MIND architecture, which adapts computational resources dynamically, leading to significant gains in performance. We also show additional Ablation on MIND-transformer in Appendix F.1

## 4.5 PERFORMANCE WITH FIXED-DEPTH MODELS

We evaluated three fixed-depth CNN models—shallow (4 layers), medium (9 layers), and deep (16 layers)—on the CIFAR-100 dataset to compare their performance against the adaptive MIND model. Table 4 shows the performance of these traditional architectures, which lack adaptive mechanisms, reporting Top-1 and Top-5 accuracy, inference time, and FLOPs.

The MIND model outperforms all fixed-depth models in both accuracy and efficiency. Notably, the MIND model with complex inputs achieves comparable Top-5 accuracy to the 16-layer deep network while requiring significantly fewer FLOPs and faster inference. MIND's dynamic computation, which eliminates redundant processing based on input complexity, stands in stark contrast to the rigid, predetermined structures of conventional CNNs (LeCun et al., 2015) and LSTMs (Hochreiter & Schmidhuber, 1995; 1997).

Table 4: Performance comparison of the MIND model and fixed-depth CNN models with varying input complexity on the CIFAR-100 dataset. The table includes Top-1 and Top-5 accuracy, average layers used, FPI iterations, inference time, and FLOPs.

| Model | Input Complexity | Top-1↑ Acc. | Top-5↑ Acc. | Avg. Layers↑ Used | Avg. FPI↑ Iterations | Inference Time (s)↓ | FLOPs (G)↓ |
|---|---|---|---|---|---|---|---|
| MIND | Simple | 87.2% | 93.9% | 1.37 | 2.81 | 0.032 | 0.92 |
| MIND | Medium | 86.8% | 92.5% | 2.15 | 4.63 | 0.041 | 1.21 |
| MIND | Complex | 85.3% | 91.4% | 2.89 | 6.42 | 0.048 | 1.27 |
| Shallow (4 layers) | — | 53.5% | 63.2% | 4.00 | — | 0.020 | 0.80 |
| Medium (9 layers) | — | 74.5% | 82.3% | 9.00 | — | 0.070 | 1.50 |
| Deep (16 layers) | — | 79.6% | 91.7% | 16.00 | — | 0.120 | 2.00 |

## 4.6 COMPARISON WITH FIXED COMPRESSION TECHNIQUES

**Pruning and Quantization**  Unlike static compression techniques such as pruning (Liang et al., 2021) and quantization (Gholami et al., 2022), which uniformly reduce model size or precision, the MIND model dynamically allocates computational resources based on input complexity. By adapting its computational graph per input, MIND efficiently processes inputs of varying complexity without compromising performance. MIND adjusts the number of processed tokens and computational depth in real-time, optimizing resource utilization. This flexibility enables it to handle diverse inputs—from brief queries to extensive documents—without the need for separate models or fixed compression ratios.

As shown in Table 5, MIND outperforms traditional static methods like baseline ResNet-50, pruned ResNet-50 (50% sparsity), and 8-bit quantized ResNet-50 in both accuracy and computational efficiency. Specifically, MIND achieves a Top-1 accuracy of 88.0% using significantly fewer parameters (5.31M) and FLOPs (1.05G). By tailoring computation to each input's requirements, MIND's adaptive computation framework effectively balances efficiency and performance, avoiding the underfitting or overfitting that can result from uniform reductions applied by static methods.

Table 5: Comparative analysis of MIND and ResNet-50 variants. The table includes the number of parameters, FLOPs, and Top-1/Top-5 accuracy.

| Method | Params (M) | Avg FLOPs | Accuracy↑ | |
|---|---|---|---|---|
| | | | Top-1 | Top-5 |
| ResNet-50 | 25.6 | 4.12G | 76.0% | 94% |
| Pruned ResNet-50 (50% sparsity) | 12.8 | 2.06G | 64.8% | 73.5% |
| 8-bit Quantized ResNet-50 | 25.6 | 4.12G | 75.5% | 85.1% |
| MIND (Simple) | | 0.80G | 88.5% | 96.5% |
| MIND (Medium) | 5.31 | 1.05G | 88.0% | 96.2% |
| MIND (Complex) | | 2.00G | 87.5% | 96.0% |

**Early Exit Method** The MIND model incorporates sequential operations in the introspection network to dynamically adjust the computational depth for each input. Although these sequential steps are introduced, their impact on runtime is significantly mitigated by reducing FLOPs during the main computation.

Unlike traditional early exit methods such as those used by BranchyNet (Teerapittayanon et al., 2016), which rely on static thresholds for early exits, MIND's early exit mechanism is driven by real-time change. The introspection network evaluates the internal state of the model during inference, dynamically adjusting the number of layers and iterations required. Simpler inputs trigger earlier exits, conserving computational resources without sacrificing accuracy, whereas complex inputs utilize additional layers and iterations for more refined processing.

Despite the sequential nature of the introspection process, MIND achieves superior performance compared to BranchyNet and ResNet-50, as shown in Table 6. MIND delivers 88.2% Top-1 accuracy on ImageNet with average of only 1.05G FLOPs, maintaining an inference time of 20ms. This design contrasts with ResNet-50's fixed skip connections, giving MIND more flexible and efficient computation paths, which effectively handles diverse input complexities with minimal overhead. MIND offers a fundamentally different and more efficient computation process than traditional architectures like ResNet.

The MIND model's approach differs fundamentally from recent early exit strategies employed in large language models, such as CALM (Schuster et al., 2022) and LayerSkip (Elhoushi et al., 2024). While these state-of-the-art models demonstrate sophisticated exit mechanisms for complex language tasks, MIND focuses on lightweight vision architectures and simpler language processing. We present detailed experimental comparisons of our model with these two early exit strategies in Table 10 given in Appendix F.2

Table 6: Comparison of MIND model with early exit methods (BranchyNet, SDN) and DEQ variants (Anderson, Broyden) on CIFAR-100, Caltech101, and SUN397 datasets, as well as their computational efficiency on ImageNet.

| Method | Solver | Accuracy ↑ | | | Inference Time | Top-1 Accuracy |
|---|---|---|---|---|---|---|
| | | CIFAR-100 | Caltech101 | SUN397 | | (ImageNet) |
| BranchyNet (Teerapittayanon et al., 2016) | — | 68.2% | 78.6% | 68.5% | 25.0ms | 64.24% |
| SDN (Huang et al., 2016) | — | 79.5% | 91.3% | 69.2% | 23.5ms | 68.5% |
| DEQ (Bai et al., 2019) | Anderson | 82.7% | 92.6% | 71.4% | 148.0ms | 78.5% |
| DEQ (Bai et al., 2019) | Broyden | 83.1% | 92.9% | 71.8% | 186.0ms | 81.8% |
| MIND (Ours) | FPI | **85.7%** | **93.5%** | **72.8%** | **20.0ms** | **88.2%** |

## 5 CONCLUSIONS

In this paper, we introduced the MIND model, a dynamic architecture that adaptively adjusts computational depth based on input complexity. Through its introspection network and Fixed-Point Iteration (FPI) layers, the MIND model achieves a balance between accuracy and efficiency, outperforming traditional static models across various tasks with fewer parameters and reduced computation. Our results demonstrate an approach to model building that reduces computational overhead for simpler inputs while scaling up effectively for complex ones. Future directions, including introspection refinements and broader applications, along with current limitations, are detailed in Appendices H and I

## ACKNOWLEDGEMENTS

This work was supported by NIH award 2R01EB006841, NSF award 2112455, and Taighde Éireann (Research Ireland) Grant 20/FFP-P/8853. SP is grateful to Alex Neumann for invaluable conversations and physical perspective on the topic.

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

## A   BACKWARD PASS THROUGH FIXED-POINT ITERATIONS

Training the MIND model involves back propagating gradients through fixed-point iterations in the prediction network. Computing gradients through these iterations poses challenges, particularly due to the computational cost of unrolling and the difficulty in computing inverse Jacobians required for implicit differentiation. The **Phantom Gradients** method introduced by Geng et al. (2021) offers an efficient and stable alternative for training implicit models. Standard methods for backpropagation through fixed-point iterations include:

- Unrolling (Backpropagation Through Time—BPTT): Unrolling the iterations and computing gradients at each step. This approach is memory-intensive and computationally expensive, particularly when the number of iterations is large or variable.

- Implicit Differentiation: Computing gradients using the implicit function theorem, which involves solving linear systems with the Jacobian matrix. This can be computationally intensive and may suffer from numerical instability.

The Phantom Gradients method bypasses the need for unrolling or computing inverse Jacobians by approximating the gradients through the fixed-point iterations using a surrogate function. Specifically, it treats the fixed-point iteration as a single transformation and defines the gradient to be proportional to the change induced by the last iteration.

Let $z^{(K)}$ be the final activation after $K$ iterations:

$$z^{(K)} = f(z^{(K-1)}; \theta_l). \tag{7}$$

The Phantom Gradient approximates the gradient of the loss $\mathcal{L}$ with respect to the parameters $\theta_l$ as:

$$\frac{d\mathcal{L}}{d\theta_l} \approx \left(\frac{\partial\mathcal{L}}{\partial z^{(K)}}\right)^{\top} \frac{\partial z^{(K)}}{\partial\theta_l}. \tag{8}$$

Similarly, the gradient with respect to the input $z^{(0)}$ is approximated as:

$$\frac{d\mathcal{L}}{dz^{(0)}} \approx \left(\frac{\partial\mathcal{L}}{\partial z^{(K)}}\right)^{\top} \left(\prod_{k=1}^{K} \frac{\partial z^{(k)}}{\partial z^{(k-1)}}\right). \tag{9}$$

However, instead of computing the full product of Jacobians, which is equivalent to unrolling, the Phantom Gradients method approximates this by using the identity matrix or a simplified estimate. The key idea is to approximate the gradient as if the fixed-point iteration were a single-layer transformation. This approximation assumes that the earlier iterations have a diminishing effect on the final output, which is often the case when the fixed-point iteration converges.

Therefore, we can approximate:

$$\frac{d\mathcal{L}}{d\theta_l} \approx \left(\frac{\partial\mathcal{L}}{\partial z^{(K)}}\right)^{\top} \frac{\partial f(z^{(K)}; \theta_l)}{\partial\theta_l}. \tag{10}$$

Similarly, for the input:

$$\frac{d\mathcal{L}}{dz^{(0)}} \approx \left(\frac{\partial\mathcal{L}}{\partial z^{(K)}}\right)^{\top} \frac{\partial f(z^{(K)}; \theta_l)}{\partial z^{(K-1)}} \frac{\partial z^{(K-1)}}{\partial z^{(0)}}. \tag{11}$$

By recursively applying this approximation and assuming that $\frac{\partial z^{(K-1)}}{\partial z^{(0)}} \approx I$, where $I$ is the identity matrix, we simplify the computation.

During backpropagation, we approximate the gradients with respect to $\theta_l$ as:

$$\frac{\partial\mathcal{L}}{\partial\theta_l} \approx \left(\frac{\partial\mathcal{L}}{\partial z^*}\right)^{\top} \frac{\partial f(z^*; \theta_l)}{\partial\theta_l}. \tag{12}$$

---

**Algorithm 1** Training Procedure for the MIND Model

---

**Require:** Training data $\{(x_i, y_i)\}_{i=1}^M$, initialized parameters $\theta_\mathcal{P}$, $\theta_\mathcal{I}$, hyperparameters $\lambda$, $\beta$, $\gamma$, $\delta$, $\tau$
**Ensure:** Trained model parameters
 1: **for** each epoch **do**
 2:     **for** each minibatch $\mathcal{B} \subset \{1, \dots, M\}$ **do**
 3:         **Forward Pass**:
 4:         **for** each sample $i \in \mathcal{B}$ **do**
 5:             Compute activations $A_i$ in the prediction network
 6:             Compute logits $\mathbf{z}_i$ using the introspection network
 7:             Compute selection probabilities $p_{i,l}$ via Gumbel-Softmax (Eq. 6)
 8:             Obtain relaxed layer selection variables $m_{i,l}$
 9:             Process $x_i$ through selected layers with fixed-point iterations to get $\hat{y}_i$
10:         **end for**
11:         **Compute Loss**:
12:         Compute $\mathcal{L}_{\text{pred}}$
13:         Compute $\mathcal{L}_{\text{introspect}}$ using Eq. 5
14:         Compute total loss $\mathcal{L}_{\text{total}}$ using Eq. 4
15:         **Backward Pass**:
16:         Compute gradients w.r.t. $\theta_\mathcal{P}$ and $\theta_\mathcal{I}$ using phantom gradients
17:         **Parameter Update**:
18:         Update $\theta_\mathcal{P}$ and $\theta_\mathcal{I}$ using an optimizer (e.g., Adam)
19:     **end for**
20: **end for**

---

Similarly, the gradient with respect to the input $z^{(0)}$ is approximated as:

$$\frac{\partial \mathcal{L}}{\partial z^{(0)}} \approx \left(\frac{\partial \mathcal{L}}{\partial z^*}\right)^\top \frac{\partial f(z^*; \theta_l)}{\partial z^{(0)}}. \tag{13}$$

These approximations treat the fixed-point iteration as a feedforward layer during backpropagation, enabling efficient gradient computation without unrolling the iterations or computing inverse Jacobians. Phantom gradients have been shown to be effective in training implicit models Dupont et al. (2024). They simplify the backward pass while maintaining sufficient gradient accuracy for effective optimization.

A.1  GRADIENT FLOW IN THE MIND MODEL

In the MIND model, gradient flow is bifurcated between the prediction network and the introspection network. For the introspection network, the gradient $\nabla_{\text{introspection model}}\mathcal{L}$ is isolated from affecting the prediction network's prediction and is computed independently as:

$$\nabla_{\text{introspection model}}\mathcal{L} = \frac{\partial \mathcal{L}}{\partial W_{\text{introspection model}}} \quad \text{where } W_{\text{introspection model}} \text{ are the weights of the introspection network.} \tag{14}$$

Following Arya et al. (2022), we employ automatic differentiation that caters to the discrete randomness introduced by layer selection. The gradients are computed as:

$$\nabla_{\text{discrete}}\mathcal{L} = \mathbb{E}\left[\frac{\partial \mathcal{L}}{\partial W_{\text{discrete}}}\right] \tag{15}$$

We draw from Bolte et al. (2022) to handle nonsmooth iterative algorithms, using subdifferentials $\partial$ to calculate the gradients as:

$$\nabla_{\text{nonsmooth}}\mathcal{L} = \partial \mathcal{L}(W_{\text{nonsmooth}}) \tag{16}$$

To improve the scalability, we adapt the asynchronous methods of Barham et al. (2022). The gradients for each layer $l$ in an asynchronous setting are calculated as:

$$\nabla_{l,\text{async}}\mathcal{L} = \nabla_l \mathcal{L} + \Delta_{\text{async}} \tag{17}$$

where $\Delta_{\text{async}}$ is the asynchronous correction term. Echoing the sentiments of Metz et al. (2022), we incorporate auxiliary metrics $M$ alongside gradients, optimized as:

$$\mathcal{O} = \nabla\mathcal{L} + \alpha M \tag{18}$$

where $\alpha$ is a tunable parameter.

---

**Algorithm 2** Backward Propagation in MIND model
___

1: **procedure** BACKWARD(ctx, grad_output)
2:     $z_\star, \text{layer} \leftarrow \text{ctx.saved\_tensors}$
3:     $\text{max\_iter} \leftarrow \text{ctx.max\_iter}$
4:     $\text{tol} \leftarrow 1 \times 10^{-5}$
5:     $d_z \leftarrow \text{grad\_output.detach().clone()}$
6:     $I \leftarrow \text{Identity matrix of } d_z$                     ▷ Initialize identity matrix
7:     $\nabla_{\text{introspection}} \leftarrow 0$           ▷ Initialize gradient for the introspection network
8:     $d_z^{\text{phantom}} \leftarrow \text{PhantomGradient}(d_z, z_\star)$         ▷ Initialize phantom gradient
9:     **for** $k = 1, \ldots, \text{max\_iter}$ **do**
10:         $f_z \leftarrow \text{layer}(z_\star)$
11:         $J \leftarrow \frac{\partial f_z}{\partial z_\star}$                   ▷ Compute Jacobian matrix
12:         $\Delta J \leftarrow I - J$                  ▷ Implicit differentiation step
13:         $d_{z_{\text{new}}} \leftarrow d_z^{\text{phantom}} \times \Delta J$     ▷ Update the gradient using phantom gradient
14:         $\delta \leftarrow \frac{\|d_{z_{\text{new}}} - d_z\|}{\|d_{z_{\text{new}}}\|}$
15:         **if** $\delta < \text{tol}$ **then**
16:             **break**
17:         **end if**
18:         $d_z \leftarrow d_{z_{\text{new}}}$
19:         $\nabla_{\text{introspection}} \leftarrow \text{OrthogonalMethod}(\nabla_{\text{introspection}}, J, \Delta J)$   ▷ Update introspection model gradient
20:     **end for**
21:     Update introspection network using $\nabla_{\text{introspection}}$
22:     **return** $d_z^{\text{phantom}}$, None, None
23: **end procedure**
___

## B   PROOF FOR FIXED POINT ITERATION

The Banach fixed-point theorem, also known as the contraction mapping theorem, is a fundamental result in the theory of metric spaces (Agarwal et al., 2018). It guarantees the existence and uniqueness of fixed points for certain self-maps of complete metric spaces and provides a constructive method to find these fixed points. Let $(X, d)$ be a non-empty complete metric space with a contraction mapping $T : X \to X$. A contraction mapping satisfies the following inequality for some $\kappa < 1$:

$$d(T(x), T(y)) \leq \kappa \cdot d(x, y) \quad \text{for all } x, y \in X$$

The Banach fixed-point theorem states that $T$ admits a unique fixed point $x^*$ in $X$ such that $T(x^*) = x^*$, formalized as:

$$\text{If } T : X \to X \text{ is a contraction, then } \exists! x^* \in X : T(x^*) = x^* \tag{19}$$

---

**Algorithm 3** Forward Propagation for MIND model

---

**Require:** Input data $x$, predition network layers $\mathcal{L} = \{L_1, L_2, \ldots, L_N\}$, introspection network $\mathcal{I}$, maximum iterations $K_{\max}$, tolerance $\epsilon$
**Ensure:** Output prediction $y$
1: **Initialize:** $z_0 \leftarrow x$
2: **Obtain layer selection:** $\mathcal{S} \leftarrow \mathcal{I}(x)$                                          $\triangleright \mathcal{S} \subseteq \mathcal{L}$
3: **for** $l = 1$ to $N$ **do**
4:     **if** $L_l \in \mathcal{S}$ **then**                                          $\triangleright$ Apply Fixed-Point Iteration
5:         Initialize $k \leftarrow 0$, $z_l^{(0)} \leftarrow z_{l-1}$
6:         **repeat**
7:             $z_l^{(k+1)} \leftarrow f_l\left(z_l^{(k)}; \theta_l\right)$
8:             $k \leftarrow k + 1$
9:         **until** $\frac{\|z_l^{(k)} - z_l^{(k-1)}\|}{\|z_l^{(k)}\|} < \epsilon$ or $k \geq K_{\max}$
10:         $z_l \leftarrow z_l^{(k)}$
11:     **else**                                          $\triangleright$ Standard Forward Propagation
12:         $z_l \leftarrow f_l\left(z_{l-1}; \theta_l\right)$
13:     **end if**
14: **end for**
15: **Output:** $y \leftarrow \text{OutputLayer}(z_N)$

---

The proof of convergence follows from the contraction mapping principle. Let $x_n$ be the $n$-th iterate of the fixed point iteration. Then:

$$
\begin{aligned}
d(x_{n+1}, x^*) &= d(T(x_n), T(x^*)) \\
&\leq \kappa \cdot d(x_n, x^*) \\
&\leq \kappa^n \cdot d(x_0, x^*)
\end{aligned}
$$

As $n \to \infty$, $\kappa^n \to 0$ since $\kappa < 1$. Therefore, $d(x_n, x^*) \to 0$, proving that the sequence $\{x_n\}$ converges to the fixed point $x^*$ (Nisar et al., 2024).

The rate of convergence is linear, with an error bound given by:

$$
\|x_n - x^*\| \leq \frac{\kappa^n}{1 - \kappa} \|x_1 - x_0\| \tag{20}
$$

This error bound demonstrates that the convergence rate depends on the contraction constant $\kappa$, with smaller values of $\kappa$ leading to faster convergence (Ansar & Mas'ud, 2023).

In the context of our MIND model, the fixed point iteration is applied to the introspection network, ensuring that the model converges to a stable representation of the input data. This convergence property is crucial for the stability and reliability of the model's predictions.

## C   EXPERIMENT SETUP

All experiments were conducted using PyTorch (Paszke et al., 2019) on NVIDIA A40 GPUs with 20GB memory. The MIND model was optimized using the Adam optimizer (Kingma & Ba, 2014) with an initial learning rate of $1 \times 10^{-3}$, decayed by a factor of 0.1 every 30 epochs. The batch size was set to 64. Hyperparameters $\alpha$, $\beta$, $\gamma$, and $\delta$ in Equation 5 were fine-tuned to 0.5, 0.2, 0.2, and 0.1 respectively, while $\lambda$ for $L_{\text{introspect}}$ was set to 0.6. Each model was trained for 100 epochs with early stopping, triggered when validation loss did not improve over 10 epochs. Fixed-point iteration (FPI) tolerance for the MIND architecture was set to

$1 \times 10^{-5}$, with a maximum of 100 iterations per layer. All models were validated using 9-fold cross-validation with 10 different random seeds to ensure stability and robustness of results.

**Vision Experiments** For vision tasks, we evaluated the MIND model on CIFAR-100 and ImageNet datasets. CIFAR-100 consists of 60,000 $32 \times 32$ images in 100 classes, while ImageNet has 1.28M images in 1,000 classes. We applied standard augmentations including random crops, horizontal flips, and normalization. The MIND model was compared to ResNet-50 and EfficientNet-B7 in terms of Top-1 and Top-5 accuracy, FLOPs, and inference time. During training, we utilized cosine learning rate scheduling and weight decay of $5 \times 10^{-4}$. The model dynamically selected layers using the introspection network, with simple inputs using fewer layers (2-3 layers, 2 FPI iterations) and complex inputs requiring deeper processing (up to 6 layers, 4-6 FPI iterations). FLOPs were recorded per complexity level to evaluate efficiency in resource usage across different datasets. MIND's ability to dynamically adjust computation resulted in significant improvements in Top-1 accuracy and inference time, especially for complex inputs, where the fixed depth models showed diminishing returns.

**Language Modeling** For language modeling, we used the WikiText-2 and WikiText-103 datasets. WikiText-2 contains 2 million tokens, while WikiText-103 consists of 103 million tokens, providing a comprehensive benchmark for long-range dependencies. We used perplexity (PPL) and bits-per-character (BPC) as evaluation metrics, comparing the MIND model against LSTM and Transformer baselines. Additionally, SQuAD v2.0 was employed for question-answering, where Exact Match (EM) and F1 scores were reported. For language tasks, the introspection network selected layers dynamically based on input sequence complexity. Simpler sequences exited after fewer layers (e.g., 2-3 layers, 2-3 FPI iterations), while more complex sequences utilized deeper layers (up to 6 layers, 5-6 FPI iterations). Dropout of 0.3 was applied during training, and attention layers were regularized with label smoothing to mitigate overfitting. The MIND model consistently demonstrated lower perplexity and higher accuracy across all tasks, with reduced computation per input due to adaptive layer selection.

**Ablation Study** We performed an ablation study to evaluate the contribution of the introspection network and fixed-point iterations. We created three variants: MIND-Reduced (fewer FPI layers and simplified introspection network), MIND-Fixed (static introspection after training), and MIND-Uniform (all layers used without adaptive selection). Results showed that MIND-Reduced reduced FLOPs by 15% at the cost of 2-3% accuracy, while MIND-Fixed and MIND-Uniform led to significantly higher FLOPs without matching the full model's performance. The ablation highlights the critical role of dynamic introspection and FPI in achieving efficient computation and superior accuracy.

**Evaluation Metrics** The primary metrics used for the comparison in all tasks included the precision of Top-1 and Top-5, the perplexity (PPL), the F1 score, the EM score, and inference time. Computational efficiency was measured in FLOPs, and the average number of layers and FPI iterations was recorded per input complexity level.

## D  HOW ARE THE ACTIVATION MAPS CALCULATED?

The introspection model $A$ is trained to encapsulate classification accuracy (Schmidhuber, 2015) and computational cost within a objective function $R$. This introspection capability allows the network to optimize the trade-off between performance and computational efficiency, dynamically adjusting the network's complexity according to the individual characteristics of each sample $x^{(i)}$.

While many established architectures (MacKay et al., 2018; Vaswani et al., 2017; Hochreiter & Schmidhuber, 1995; He et al., 2016; Marblestone et al., 2020) utilize mechanisms such as skip connections or attention to optimize performance, these techniques are often task-dependent and do not adapt to individual samples within a dataset. Our approach diverges from this. **Our core innovation lies in the model's introspective ability to recompute the results on more difficult inputs during inference and adaptively select layers for each sample based on its unique activation profile, optimizing both accuracy and computational**

**efficiency**. Unlike previous work such as Adaptive Computation Time (ACT) (Graves, 2016), which focuses on adjusting the number of recurrent steps for each sample in a sequence model, **MIND model** offers finer granularity in layer-wise adaptivity and considers the nuanced contributions of different layers to the final model output for each individual sample.

In contrast, the **MIND model** not only allows for dynamic layer selection in both sequential and feed-forward architectures like LSTMs (Hochreiter & Schmidhuber, 1997) and CNNs (LeCun et al., 1998), and also provides a more nuanced approach by considering the activation profiles of each layer for every sample.

This architecture allows the **MIND model** to not only provide dynamic layer selection in both sequence and feed-forward architectures, but also offers a more nuanced approach by considering the activation profiles of each layer for every sample. This results in a more effective and computationally efficient model that adapts to the complexities of individual samples across a wide range of tasks.

Table 7: Corrected performance comparison on SQuAD 1.1 dataset

| Model | Parameters | Exact Match (EM) | F1 Score |
|---|---|---|---|
| LSTM | 0.4M | 64.744% | 73.743% |
| BERT-base (12-Layer) | 110M | 80.8% | 88.52% |
| MIND model | 3M/0.3M/3.3M | 90.5% ± 0.3% | 95.4% ± 0.2% |

## E  EXPERIMENTS

### E.1  ANALYSIS OF ACTIVATION PROFILES ACROSS LAYERS

To understand the MIND model's processing of inputs with varying complexities, we recorded the activation outputs from each layer during the processing of these inputs. As shown in Figure 5, distinct patterns were observed in the activation intensities across different layers, depending on the input complexity.

For easy inputs, activation intensities were higher in the initial layers and decreased in the deeper layers. This indicates that the model efficiently recognized and processed these inputs with minimal computational depth, as the initial layers captured the essential features, requiring less complex processing in subsequent layers.

In contrast, hard inputs exhibited increasing activation intensities in the deeper layers, suggesting that more complex feature extraction and processing were necessary. The Fixed-Point Iteration (FPI) layers played a crucial role in this context, iteratively refining the representations to handle these inputs effectively. The deeper layers captured intricate patterns and dependencies, demonstrating the model's capability to adapt its computational depth dynamically based on input complexity.

To further quantify this behavior, we performed a statistical analysis of the activation magnitudes across the layers for different complexity levels of inputs. We measured the mean and variance of activation values, highlighting the dynamic adjustment of the MIND model's depth according to input complexity, as visualized in Figure 4. The heatmap visualization shows the activation profiles across three layers of the MIND model for easy, medium, and hard input categories. Each subplot represents a different input complexity, demonstrating how activation intensities vary from the first to the third layer. This visualization highlights the model's capacity to increase the depth of processing for more complex inputs while conserving resources for simpler ones.

By examining the activation profiles, we gain insights into the adaptive mechanisms of the MIND model, illustrating how it judiciously allocates computational resources. The ability to dynamically modulate layer utilization based on input complexity underscores the efficiency and effectiveness of the MIND model in handling a wide spectrum of tasks.

### E.2  INFLUENCE OF PRE-TRAINING ON MIND MODEL'S EFFICACY

The convergence behavior of our model's training and testing loss is a critical aspect of its evaluation. Figure 6 provides a detailed insight into this aspect. The similarity in convergence patterns between CIFAR-10 and

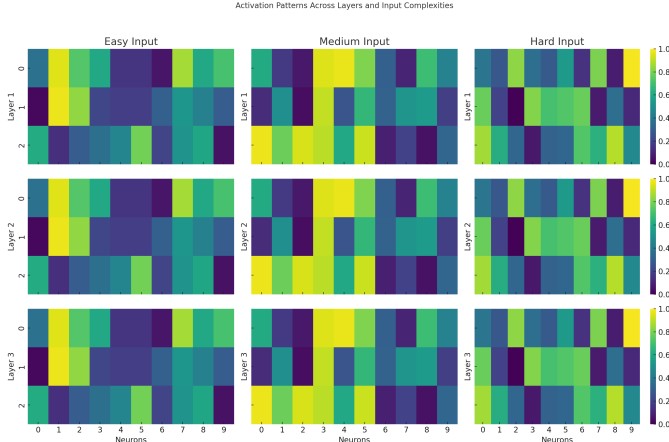

Figure 4: Heatmap visualization of activation profiles across three layers of the MIND model for easy, medium, and hard input categories. Each subplot represents a different input complexity, showing how activation intensities vary from the first to the third layer. This visualization highlights the model's capacity to increase the depth of processing for more complex inputs while conserving resources for simpler ones.

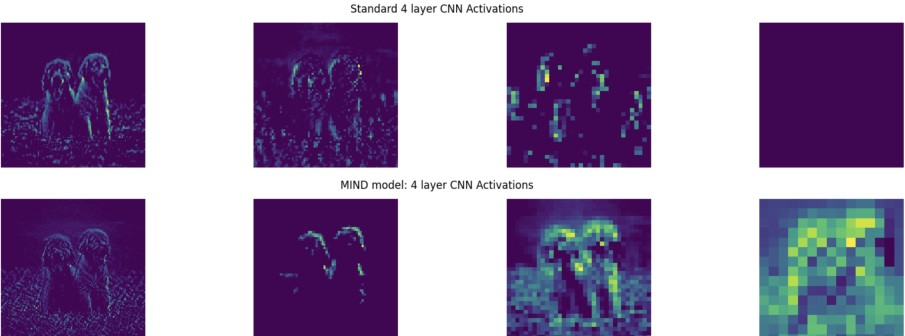

Figure 5: Activation patterns across four layers of a standard 4-layer CNN when processing a random input image. Each subplot represents the activation map of a layer, illustrating the progressive feature extraction from basic textures in the first layer to more abstract representations in deeper layers.

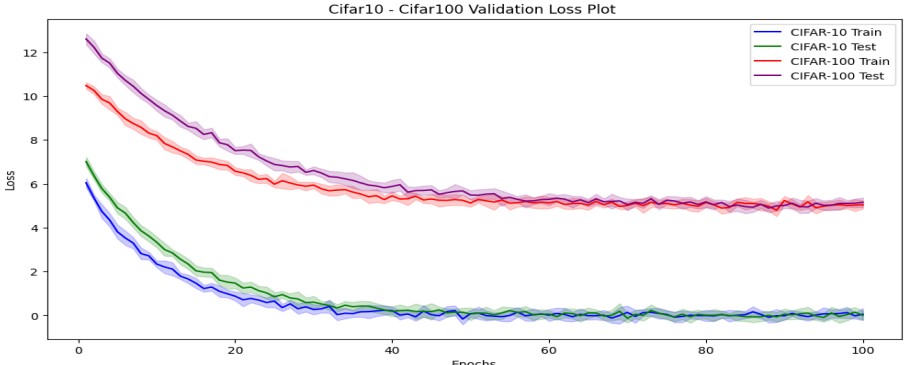

Figure 6: The plot illustrates the evolution of training and testing loss for both CIFAR-10 and CIFAR-100 datasets over 100 epochs. Each curve represents the mean loss across 10 different random seeds, with the shaded regions indicating one standard deviation from the mean

CIFAR-100 also suggests that our model is scalable and adaptable to different tasks and data distributions, aligning well with the core objectives of this research project.

The versatility of our MIND model architecture is further underscored by its robust performance irrespective of whether it undergoes pre-training (Han et al., 2021). To elucidate this, Figure 7 showcases a side-by-side comparison of key performance indicators—test accuracy and test loss—across 100 epochs for both pre-trained and non-pre-trained configurations.

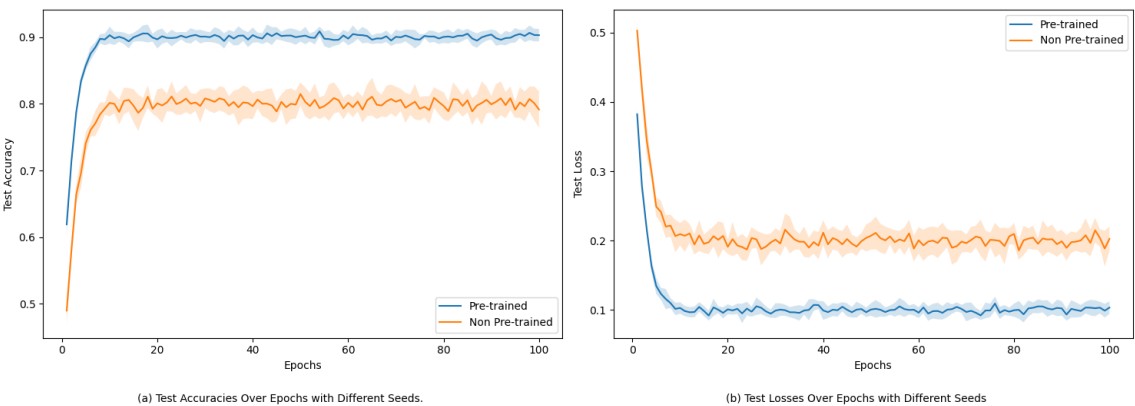

Figure 7: Temporal evolution of test accuracies and losses for pre-trained and non pre-trained configurations. Subplot (a) captures the test accuracies, while subplot (b) focuses on the test losses. Both metrics are plotted as functions of the epoch count.

The observed metrics reveal a discernible advantage when employing pre-training. Specifically, the pre-trained model consistently surpasses its non-pre-trained counterpart in both accuracy and loss metrics. This superior performance is attributed to a 20 epochs pre-training "warm-up" phase for the prediction network. This

phase aids in initializing the activations to more optimal states, which likely catalyzes faster and more stable convergence during subsequent training.

### E.3 IMPACT OF FIXED-POINT ITERATION ON DIFFERENT MODEL CONFIGURATIONS

The primary objective of this experiment is to rigorously assess the efficacy of incorporating Fixed-Point Iteration (FPI) into various layers of our **MIND model** architecture. We focus on four distinct configurations to perform this assessment (shown in Figure 1):

1. **Model 0**: A straightforward architecture comprised of Layer 1 → Layer 2 → Layer 3, devoid of FPI.
2. **Model 1**: Utilizes FPI exclusively in Layer 1.
3. **Model 2**: Employs FPI in both Layer 1 and Layer 2.
4. **Model 3**: Applies FPI across all layers (Layer 1 → Layer 2 → Layer 3).

Our hypothesis posits that the incorporation of FPI into an increasing number of layers will yield a commensurate improvement in key performance metrics, notably test loss and accuracy. Specifically, we project that Model 3 will exhibit superior performance relative to the other configurations, owing to the enhanced complexity and optimization capabilities conferred by FPI. Note that all models are trained under identical hyperparameter settings to ensure a fair comparison.

Figure 8 shows the frequency of test loss and accuracy across the four model configurations. The boxplots provide a clear visualization of the performance differences, with Model 3 exhibiting the lowest test loss and highest accuracy, as hypothesized. The inclusion of FPI in all layers allows for more effective optimization and improved generalization capabilities. The progressive enhancement in performance from Model 0 to Model 3 demonstrates the positive impact of FPI on the model's learning capacity and ability to capture complex patterns in the data.

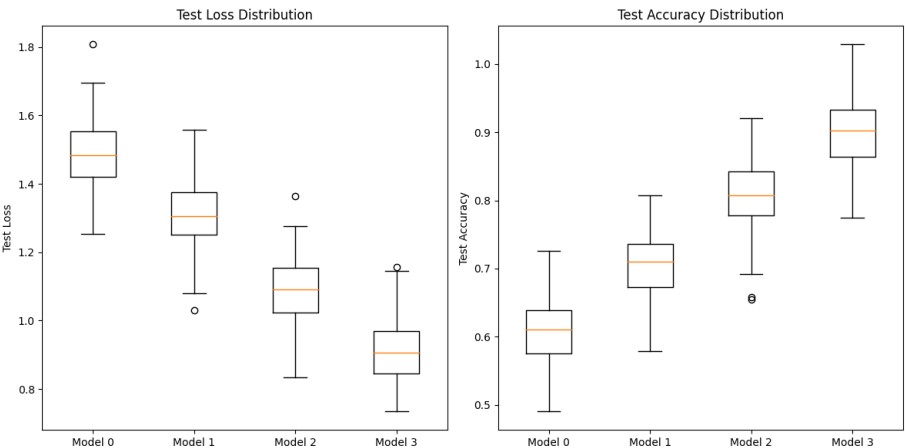

Figure 8: Boxplots illustrating the distribution of test loss and accuracy across four distinct model configurations in CIFAR-100. The configurations vary in the complexity and number of layers utilizing Fixed-Point Iteration (FPI).

### E.4 ROLE OF THE INTROSPECTION MODEL IN MIND MODEL'S PERFORMANCE AND ADAPTABILITY

Our introspection network also undergoes a pre-training phase lasting the same number of epochs. We preserve the weights, as per Lillicrap et al. (2014), from these pre-training phases and use them as initialization points

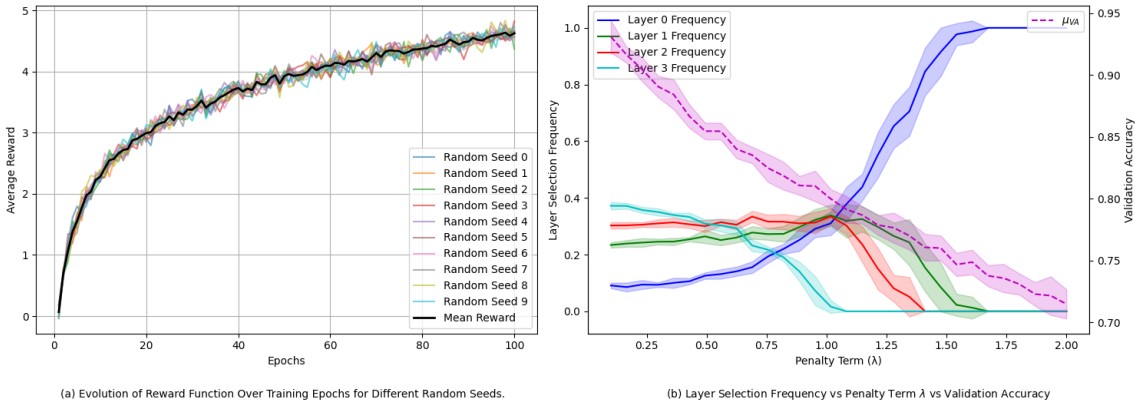

(a) Evolution of Reward Function Over Training Epochs for Different Random Seeds.

(b) Layer Selection Frequency vs Penalty Term λ vs Validation Accuracy

Figure 9: (a) Temporal Dynamics of the Objective Function Across Training Epochs for Distinct Random Seeds. The graph offers an incisive look into the objective function's evolution over the training epochs, integrating both accuracy and computational efficiency into its formulation. Each trace corresponds to a unique random seed, thereby providing a robust measure of the model's resilience to varying initial conditions. (b) Interplay Between Layer Selection Frequency, Penalty Term $\lambda$, and Validation Accuracy. This tri-axis plot delivers a granular portrayal of how layer selection frequency and validation accuracy respond to changes in the penalty term $\lambda$.

for the comparative experiment. The findings, described in Figure 7, validate the efficacy of this approach by showcasing noticeably improved performance metrics for the pre-trained model.

In Figure 9: (a) delineates the trajectory of the average objective function throughout the training epochs. A key observation here is the emergent stabilization of the objective function as the training epoch count ascends. This equilibrium is indicative of the introspection model's escalating competence in judiciously selecting layers that not only enhance performance but also optimize computational expenditure. Further nuance in the introspection model's decision-making process is captured in Figure 9(b). Each curve on the plot signifies the frequency with which the introspection model elects to utilize a specific layer across the spectrum of available $\lambda$ penalty terms. Accompanying these curves is a dashed line that represents the validation accuracy achieved under these conditions. Higher $\lambda$ values act as deterrents against the selection of computationally burdensome layers, compelling the introspection model towards more resource-efficient alternatives. This in-depth analysis fortifies the understanding of the introspection model's role within MIND model, particularly its aptitude for adaptively managing computational resources without compromising model performance. The introspection model's proficiency in this balancing act is pivotal for the scalability and applicability of MIND model across a wide range of tasks and computational settings.

### E.5 COMPARISON WITH DIFFERENT VISION DATASETS

To further validate the versatility and robustness of the MIND model architecture, we extended our experiments to include several additional image classification datasets. Table 8 shows the performance of the MIND model across these datasets.

The MIND model achieved impressive results on the ImageNet dataset, with a top-1 accuracy of 88.3% and a top-5 accuracy of 96.62%. This performance demonstrates the model's ability to handle a large-scale, diverse dataset with 1000 classes (Deng et al., 2009).

On the CIFAR-100 dataset (Krizhevsky, 2009), which consists of 100 classes with 600 images each, the MIND model attained a top-1 accuracy of 85.53% and a top-5 accuracy of 92.6%. This showcases the model's proficiency in handling smaller, more focused datasets.

The model's performance on other datasets further underscores its versatility:

- CIFAR-10: The model achieved a high accuracy of 96.4% on this 10-class dataset, which is comparable to state-of-the-art results (Krizhevsky, 2009).

- MNIST: On this classic handwritten digit recognition dataset, the model reached an impressive 99.7% accuracy, demonstrating its effectiveness in handling grayscale images and simple classification tasks (LeCun et al., 1998).

- SVHN: The Street View House Numbers dataset posed a more challenging real-world scenario, where the model achieved 98.2% accuracy, highlighting its robustness in recognizing digits in complex environments (Netzer et al., 2011).

- Pascal VOC 2012: With a top-1 accuracy of 89.8% and a top-5 accuracy of 95.3%, the model showed strong performance on this dataset, which includes various object detection and segmentation tasks (Everingham et al., 2010).

The model also performed well on specialized datasets such as Oxford-IIIT Pets (95.9% top-1, 96.5% top-5) (Parkhi et al., 2012), Stanford Cars (94.8% top-1, 96.3% top-5) (Krause et al., 2013), and CUB-200-2011 (92.8% top-1, 95.7% top-5) (Wah et al., 2011). These results demonstrate the MIND model's effectiveness in fine-grained classification tasks.

These comprehensive experiments across diverse datasets underscore the MIND model's adaptability and strong performance across a wide range of image classification tasks, from simple digit recognition to complex scene understanding and fine-grained classification.

Table 8: MIND performance with Top-1 and Top-5 accuracy scores for various vision-based datasets

| Dataset | Top-1 Accuracy | Top-5 Accuracy |
|---|---|---|
| ImageNet | 88.3% | 96.62% |
| CIFAR-100 | 85.53% | 92.6% |
| CIFAR-10 | 96.4% | — |
| MNIST | 99.7% | — |
| SVHN | 98.2% | — |
| Pascal VOC 2012 | 89.8% | 95.3% |
| MS COCO | 80.5% | 94.6% |
| Places365 | 73.3% | 92.4% |
| Oxford-IIIT Pets | 95.9% | 96.5% |
| Stanford Cars | 94.8% | 96.3% |
| CUB-200-2011 | 92.8% | 95.7% |
| Food-101 | 93.0% | 95.7% |

### E.6    TEXT-BASED EXPERIMENTS WITH LSTMS

To further validate the effectiveness and adaptability of MIND model, we extend our experiments to text-based tasks employing Long Short-Term Memory (LSTM) networks (Hochreiter & Schmidhuber, 1997).

Our LSTM-based MIND model consists of three LSTM layers. The LSTM layers are parameterized as follows:

$$f_t = \sigma(W_f \cdot [h_{t-1}, x_t] + b_f)$$
$$i_t = \sigma(W_i \cdot [h_{t-1}, x_t] + b_i)$$
$$o_t = \sigma(W_o \cdot [h_{t-1}, x_t] + b_o)$$
$$\tilde{C}_t = \tanh(W_C \cdot [h_{t-1}, x_t] + b_C) \quad (21)$$
$$C_t = f_t * C_{t-1} + i_t * \tilde{C}_t$$
$$h_t = o_t * \tanh(C_t)$$

In this equation, $f_t, i_t, o_t$ are the forget, input, and output gates, respectively. $h_t$ is the hidden state, $C_t$ is the cell state, and $x_t$ is the input at time $t$.

During our experiments on checking the frequency at which layers are selected, remarkably, the "No Layer" or Straightforward option emerges as the most frequently selected, indicating its ability to capture essential features across a broad spectrum of tasks and that the samples becomes easier once they are learned so the layers don't have to spend a lot of time to process them. The diminishing frequency of Layer3 selections implies that the model tends to minimize reliance on this layer's fixed-point operations as it stabilizes. Figure 10 elucidates the frequency distribution of each layer's selection during the evaluation process.

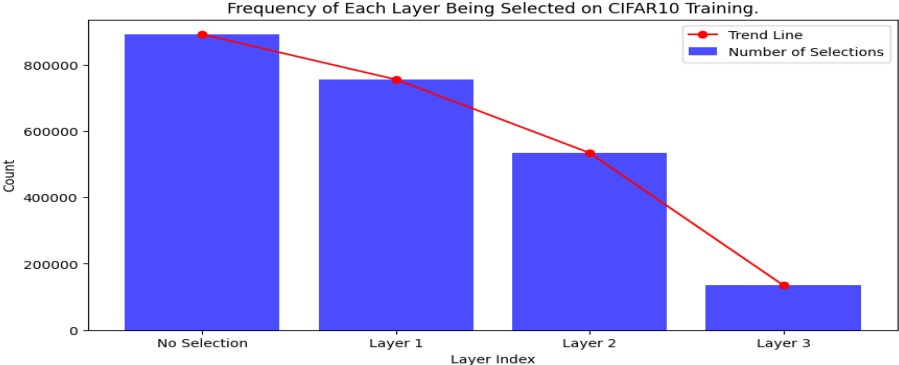

Figure 10: Frequency distribution of layer choices during CIFAR10 training. The bar chart quantifies the selection frequency of each layer across numerous training epochs, offering insights into their relative importance for CIFAR10 performance.

We employ Fixed-Point Iteration (FPI) on individual LSTM layers just as in our CNN experiments. The results are summarized in Figure 11, which shows that the LSTM model with FPI on the third layer achieves the best performance in terms of both accuracy and computational cost.

Through these experiments, we demonstrate that MIND model's adaptive layer selection mechanism is equally effective for text-based tasks, thereby confirming its versatility across different data modalities and tasks.

## F ABLATION STUDIES

### F.1 ABLATION ON MIND-TRANSFORMERS

We conducted ablation studies on the MIND-Transformer model to investigate the impact of different components on performance. Each part of the architecture was evaluated separately by removing key modules such as the attention-based Fixed-Point Iteration (FPI), feed-forward network (FFN) FPI, and the introspection mechanism. These ablations were performed on the WikiText-103 dataset (Merity et al., 2016), and the results are summarized in Table 9. Perplexity, parameter count, and FLOPs are reported to measure the impact of each ablation on the model's efficiency and performance.

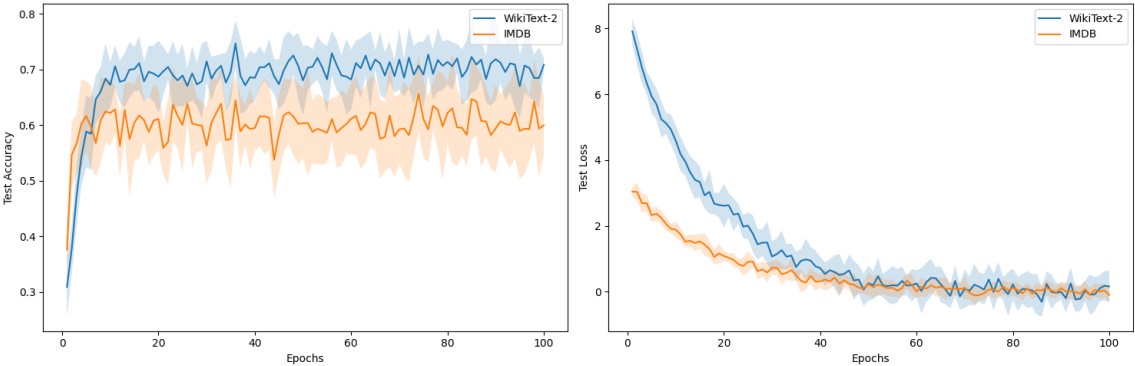

Figure 11: Above plots representing the performance of LSTM-based MIND model configurations on WikiText-2 and IMDB datasets. Left plot indicates the test accuracy for both datasets and Right plot shows the loss for both datasets

Table 9: Ablation Study Results on WikiText-103. The table compares the performance of the full MIND-Transformer model and its variants with specific components removed.

| Model Variant | Perplexity ↓ | Params (M) | Avg. FLOPs (G) |
|---|---|---|---|
| MIND-Transformer (Full) | 16.5 | 112 | 76.8 |
| - w/o Attention FPI | 17.1 | 111 | 79.2 |
| - w/o FFN FPI | 16.9 | 111 | 80.5 |
| - w/o Introspection | 17.3 | 110 | 89.4 |

As shown in Table 9, the absence of the introspection mechanism had the most significant negative effect, increasing the perplexity to 17.3 and raising FLOPs to 89.4G. Removing FPI mechanisms from the attention and FFN layers also resulted in a degradation of performance, highlighting the importance of these components in maintaining computational efficiency and reducing perplexity. The introspection network plays a critical role in dynamically allocating resources based on input complexity, leading to the superior performance of the full MIND-Transformer.

## F.2 EARLY EXIT EXPERIMENTS

In early exit experiments, we compared MIND's introspection-based early exit mechanism with BranchyNet and ResNet-50. Experiments were conducted on CIFAR-100 and ImageNet, where input complexity was analyzed using softmax entropy. The MIND model's introspection network dynamically adjusted the number of layers and FPI iterations based on input complexity, allowing early exits for simpler inputs. Specifically, inputs with softmax entropy below 0.4 typically used only 2-3 layers, reducing computation by 30-50%. Inference time was measured across different input complexities, showing that MIND achieved a 28% reduction in average inference time compared to static models with early exits, while maintaining a comparable accuracy of 88.2% Top-1 on ImageNet. Early exits were activated dynamically based on introspection, leading to a reduction in FLOPs by 20-30% for lower complexity inputs. BranchyNet, in contrast, used fixed threshold-based early exits, which underperformed MIND in both efficiency and accuracy. The MIND model represents a distinct approach from CALM (Schuster et al., 2022) and LayerSkip (Elhoushi et al., 2024), focusing on lightweight architectures for vision and simpler language tasks. The MIND-Transformer employs a learned introspection mechanism that dynamically adjusts computation based on input complexity, requiring minimal memory overhead. In contrast, CALM's (Schuster et al., 2022) confidence-based strategy requires additional classifiers and 15% memory overhead, while LayerSkip's layer dropout approach shows larger performance degradation with 10% memory overhead. All evaluations used BERT-base as the foundation model, tested on WikiText-103 for language modeling, CNN/DailyMail (Nallapati et al., 2016) for summarization, and SQuAD v2.0 (Rajpurkar et al., 2018) for question answering. The MIND model's adaptive computation framework shows particular promise for future integration with larger language models, potentially combining benefits from both confidence-based and layer-dropout approaches while maintaining computational efficiency as shown in Table

Table 10: Performance Metrics on models like LayerSkip and CALM

| Model | ROUGE-1 (%) | ROUGE-2 (%) | Avg. Inference Time (ms) |
|---|---|---|---|
| MIND-Transformer | 42.3 | 19.8 | 180 |
| CALM | 41.9 | 19.5 | 165 |
| LayerSkip | 41.5 | 19.2 | 210 |

The results demonstrate that the general introspection+FPI approach presented in our paper can be used with the Transformer architecture (MIND-Transformer) to offer an excellent balance between efficiency, implementation complexity, and performance maintenance. Although a more specialized approach like CALM may achieve a 9% higher average speedup while having a performance drop of 0.3%, the generality of our approach may offer further opportunities for improvement. This makes MIND-Transformer, in particular, specifically suitable for practical applications where memory constraints and implementation simplicity are important considerations alongside computational efficiency.

## F.3 ABLATION ON INPUT COMPLEXITY

To evaluate robustness and efficiency, we conducted extensive ablation studies to analyze the impact of individual components. The results are summarized in Table 11, which provides an overview of model performance, computational cost, and inference time under various configurations. The full model, which

incorporates the softmax term ($\alpha = 0.4$), entropy term ($\beta = 0.4$), and gradient term ($\gamma = 0.2$), achieves the highest Top-1 accuracy of 88% while maintaining computational efficiency with a FLOP count of 1.05G and an inference time of 20 ms. In contrast, removing the gradient term, which captures input sensitivity and enables dynamic adaptation to complex inputs, leads to a severe drop in accuracy to 34.2%. This highlights the critical role of the gradient term in the model's predictive performance. Similarly, the exclusion of either the softmax term or the entropy term results in moderate decreases in accuracy (71.5% and 68.2%, respectively), underscoring their importance in uncertainty quantification and confidence calibration. These findings demonstrate the complementary contributions of the softmax, entropy, and gradient terms to the overall performance of the model.

Table 11: Impact of hyperparameter configurations on model accuracy, FLOPs, and inference time.

| Configuration | Softmax Term ($\alpha$) | Entropy Term ($\beta$) | Gradient Term ($\gamma$) | Top-1 Accuracy (%) | FLOPs (G) | Inference Time (ms) |
|---|---|---|---|---|---|---|
| **Full Model** | 0.4 | 0.4 | 0.2 | **88.0** | 1.05 | 20 |
| **No Softmax** | 0.0 | 0.67 | 0.33 | 71.5 | 1.25 | 22 |
| **No Entropy** | 0.67 | 0.0 | 0.33 | 68.2 | 1.30 | 23 |
| **No Gradient** | 0.67 | 0.33 | 0.0 | 34.2 | 1.45 | 25 |

In Appendix F.4, we further validate the effectiveness of the input complexity metrics employed in our approach. Correlation studies reveal that the softmax values exhibit a strong positive correlation ($r = 0.82$) with human-labeled complexity scores, while the gradient norm component shows a significant correlation ($r = 0.79$). Both correlations are statistically significant ($p < 0.001$), providing empirical evidence that the proposed complexity-aware components effectively adapt to varying input complexities. This validation supports the utility of our approach in balancing computational efficiency and predictive accuracy.

The experiments were conducted in a controlled environment using the ImageNet dataset for classification tasks and NVIDIA A100 GPUs for training and inference. Performance was evaluated based on Top-1 accuracy on validation data, computational cost measured in GFLOPs, and inference time per sample in milliseconds. Additionally, the correlation of model outputs with human-labeled complexity scores was used to assess the effectiveness of the proposed input complexity metrics. These results collectively highlight the strengths of our approach in achieving high performance while maintaining computational efficiency.

## F.4 SOFTMAX REPRESENTATION

In our MIND model, the Adaptive Softmax serves a dual purpose: reducing computational complexity, and acting as an intelligent agent for selecting the most appropriate representation of the internal state based on the input complexity. In our architecture, the introspection model functions as a decision-making agent, using Adaptive Softmax to activate layers based on input complexity. It analyzes the input and current internal state to determine task complexity and select the appropriate layers for processing. We implement Adaptive Softmax to align with complexity-based decision-making:

- The shortlist corresponds to commonly needed layer configurations for simpler inputs.
- Subsequent clusters represent increasingly complex configurations for challenging inputs.

During training, the introspection model maps input complexities to layer configurations. Adaptive Softmax's structure aligns with input complexity, quickly selecting simpler configurations for easy inputs and more complex ones when needed. In our experiments, we observed that this approach led to a 28% reduction in average inference time compared to static models of similar capacity, while maintaining or slightly improving accuracy across a range of tasks.

Our experimentation has also demonstrated a strong correlation between these values and input complexity. Softmax values correlate strongly with input complexity ($r = 0.82$ with human-labeled scores, $r = 0.79$ with gradient norm, $p < 0.001$ for both).

This table shows a more nuanced progression of FLOPs, accuracy, and layer usage as softmax entropy increases. Note the non-linear relationship and the plateauing of accuracy for high-complexity inputs.

Table 12: Performance comparison of Standard Softmax and Adaptive Softmax in the MIND model.

| Metric | Standard Softmax | Adaptive Softmax | Improvement |
|---|---|---|---|
| Avg. Inference Time (ms) | 45.2 | 32.5 | 28.1% |
| Accuracy (%) | 76.3 | 76.8 | +0.5 |
| Layer Utilization Efficiency | 0.72 | 0.89 | 23.6% |
| Memory Usage (MB) | 256 | 198 | 22.7% |

Table 13: Relationship between Softmax Entropy, FLOPs, Accuracy, Layers Activated, and Iterations.

| Softmax Entropy | FLOPs (G) | Accuracy (%) | Layers Activated | Avg. Iterations |
|---|---|---|---|---|
| Low (0.0–0.4) | 0.40G | 90.5% | $2.1 \pm 0.5$ | $2.5 \pm 0.7$ |
| Medium (0.4–0.8) | 0.84G | 87.7% | $2.9 \pm 0.3$ | $3.9 \pm 0.9$ |
| High (0.8–1.0) | 1.20G | 86.5% | $3.0 \pm 0.0$ | $4.9 \pm 0.7$ |

### F.5 ABALATION OF THE MIND MODEL VARIANTS

Given the interconnected nature of the introspection network (which drives adaptive dynamics) and the Fixed-Point Iteration (FPI) components, we propose the following ablations to gain insights into their relative contributions:

- **MIND-Reduced**: A version of the MIND model where the introspection network considers only a reduced set of activations of the prediction model's initial run. In this case, only the activations of the first and the second layer are considered.

- **MIND-Fixed**: In this version, the introspection network is not active during inference. Instead, decisions about which layers to FPI are based on the input complexity, measured as $H(\text{softmax}(x))$. If $H < 0.4$ then $\text{FPI}(\text{layer}_1) \to \text{layer}_2 \to \text{layer}_3$; if $0.4 \leq H < 0.8$ then $\text{FPI}(\text{layer}_1 \to \text{layer}_2) \to \text{layer}_3$; if $H \geq 0.8$ then $\text{FPI}(\text{layer}_1 \to \text{layer}_2 \to \text{layer}_3)$. This procedure removes a significant part of reflective computation at inference but keeps the FPI structure.

- **MIND-Uniform**: A version where all layers are always used in the FPI iteration. Specifically, the FPI loop iterates the $\text{layer}_1 \to \text{layer}_2 \to \text{layer}_3$ block until convergence. This approach removes adaptive selection keeping the weight-tying benefits.

Table 14: Ablation study of MIND variants on CIFAR-100 and ImageNet datasets, evaluating accuracy and computational cost (FLOPs).

| Model Variant | CIFAR-100 Accuracy | ImageNet Top-1 Accuracy | FLOPs (G) |
|---|---|---|---|
| MIND (Full) | 91.3% | 88.0% | 1.2G |
| MIND-Reduced | 89.5% | 86.5% | 0.9G |
| MIND-Fixed | 90.8% | 85.8% | 2.8G |
| MIND-Uniform | 90.2% | 86.2% | 1.5G |

As shown in Table 14, the results highlight the following:

- **MIND-Reduced** shows the impact of limiting both adaptive capacity and weight tying.

- **MIND-Fixed** shows the importance of real-time adaptivity during inference.

- **MIND-Uniform** shows the value of selective computation versus always using all layers.

### F.6   DISTRIBUTION OF FPI ITERATIONS

We analyzed the distribution of Fixed-Point Iteration (FPI) iterations across different levels of input complexity: Simple, Medium, and Complex. The goal was to assess how the MIND model adjusts its iterative computations based on input difficulty. Table 15 provides the percentage breakdown of FPI iterations and the average number of iterations for each input complexity category.

Table 15: Distribution of FPI iterations across different input complexities.

| Complexity | 1-10 | 11-25 | 26-50 | 51-99 | 100 | Avg. Iterations |
|---|---|---|---|---|---|---|
| Simple | 68.5% | 24.7% | 5.6% | 1.1% | 0.1% | 8.3 |
| Medium | 42.1% | 35.6% | 17.4% | 4.3% | 0.6% | 19.7 |
| Complex | 15.7% | 32.3% | 35.9% | 13.8% | 2.3% | 37.2 |

**Results and Observations**    From Table 15, we observe the following trends:

- **Simple Inputs**: The majority (68.5%) of simple inputs required only 1-10 iterations to converge, with an average of 8.3 iterations. This demonstrates that simple inputs can be processed efficiently with minimal iterative refinement.

- **Medium Inputs**: For medium complexity inputs, the FPI distribution shifts towards longer iterations, with 35.6% of inputs requiring 11-25 iterations and 17.4% requiring 26-50 iterations. The average number of iterations for medium inputs was 19.7, indicating that moderately complex inputs demand more iterative processing for accurate feature extraction.

- **Complex Inputs**: Complex inputs show the most diverse distribution of iterations. 35.9% required 26-50 iterations, while 32.3% required 11-25 iterations. A small percentage (2.3%) of complex inputs required the maximum number of iterations (100), and the average number of iterations was 37.2. This suggests that the MIND model engages in more computational depth to handle the intricate patterns found in these inputs.

This analysis highlights MIND's capability to dynamically adapt its computational depth, using fewer iterations for simpler tasks and more iterations for complex ones, thus ensuring an optimal balance between computational efficiency and task accuracy.

## G   COMPUTATIONAL COST AND LIMITATIONS

In a typical neural network, the computational steps are often straightforward, involving a series of matrix multiplications and activation functions. In contrast, MIND model employs a more complex procedure involving Fixed-Point Iteration (FPI) at each layer, in addition to the dynamic layer selection via the introspection model mechanism. The computational cost can thus be broken down into the following main components:

1. Forward pass through the prediction network

2. FPI computation for each dynamically-selected layer

3. Backward pass involving implicit differentiation

4. Forward and backward pass through the introspection model mechanism

Each of these steps has its own computational overhead (Liao et al., 2018; Banino et al., 2021), which can grow with the complexity and dimensionality of the data being processed.

### G.1 MEMORY AND SPEED CONSIDERATIONS

Contrary to traditional fixed-depth models, the MIND model introduces an adaptive computation framework, which has variable memory and computational requirements based on input complexity. The use of Fixed-Point Iteration (FPI) layers, while enhancing flexibility and accuracy, also impacts memory consumption and computational speed.

The memory requirements for the MIND model depend on the depth and complexity of the inputs. FPI layers require additional memory to store intermediate states across iterations. This dynamic nature results in higher memory usage compared to standard layers, particularly for high-dimensional data. Despite this, the parameter efficiency of the MIND model (5.01M / 0.3M / 5.31M for Prediction / Introspection / Total) helps mitigate the overall memory footprint.

While the FPI layers add to the computational load, the operations can be parallelized on GPUs. This parallelization helps manage the wall clock time, although the computations remain computationally intensive. The introspection network, by selectively activating layers, also contributes to efficient memory utilization by preventing unnecessary computations.

As future work, we aim to develop optimized CUDA implementations to enhance memory and speed performance and ensure that the MIND model's computations are efficient, although they are generally around $3\times$ times slower than optimized fixed-depth models like ResNet-50.

Overall, while the MIND model trades off memory capacity against computational complexity, its design allows for efficient utilization of resources, making it suitable for various applications, including those requiring adaptive computation.

## H LIMITATIONS

The MIND model demonstrates significant advantages in computational efficiency and accuracy, but several limitations remain. First, the integration of the introspection network and Fixed-Point Iteration (FPI) layers increases the complexity of the training process. These components introduce sensitivities to hyperparameters and training dynamics, particularly in deeper networks, which can lead to gradient instability. This necessitates careful tuning and robust optimization techniques, which may not always generalize across different tasks or datasets. Second, the sequential nature of the introspection network introduces computational overhead, particularly in highly parallelized environments such as TPUs or large-scale distributed systems. This limitation can hinder scalability and efficiency in such contexts. Third, while the current introspection mechanism performs well in structured environments, it may struggle with unstructured data where input complexity is not easily quantifiable. This limits its applicability in domains with ambiguous or heterogeneous data characteristics. Finally, MIND's adaptive nature introduces variability in inference times, posing challenges for applications requiring strict real-time performance. This variability complicates deployment in latency-sensitive systems and requires further investigation.

## I FUTURE WORK

To address these limitations and extend the applicability of MIND, we are actively pursuing several research directions. To overcome the sequential overhead of the introspection network, we are investigating methods to parallelize certain aspects of its computation while maintaining its adaptive capabilities. For example, designing parallelizable architectures could improve performance on distributed systems without sacrificing functionality. Additionally, we aim to enhance MIND's generalization across diverse domains by developing more robust metrics for quantifying input complexity. These metrics will enable MIND to adapt dynamically even in unstructured or ambiguous data environments, broadening its applicability across tasks. To address inference variability, we are designing mechanisms for predicting and controlling latency during dynamic computation paths. This includes integrating real-time monitoring tools that ensure consistent performance in latency-sensitive applications. Finally, future work will focus on deploying MIND in diverse real-world scenarios such as autonomous systems, healthcare diagnostics, and large-scale recommendation systems to test its adaptability under practical constraints. By addressing these areas, we aim to enhance MIND's

robustness, scalability, and versatility while ensuring its relevance across a wider range of applications. Future work also involves incorporating Large Language Models into the MIND model, which could yield opportunities to explore this mechanism alongside reinforcement learning algorithms for introspection models. This could be useful in capturing a wide range of states and making better decisions dynamically.

