# OpenReview forum: "MIND over Body: Adaptive Thinking using Dynamic Computation"
_ICLR.cc/2025/Conference — ICLR 2025 Oral_

### Official Review · Reviewer_P7eZ · 2024-11-01

**Soundness:** 3
**Presentation:** 3
**Contribution:** 3
**Rating:** 8
**Confidence:** 4

**Summary:**

The authors propose a framework for designing architectures that are able to adaptively control the number of computations they perform in order to produce an output given an input. Their proposal has two main components. First, separate the model into two parts: a prediction network and a control network. At each step, the latter decides which layers to use to refine it’s estimate of the output. To make computations even more granular, they use a Deep Equilibrium networks to implement each layer as a fixed-point iteration computation. Thus the control network can no only decide which layers to apply, but for how long. The authors proceed to show the effectiveness of their approach on several qualitatively different datasets.

**Strengths:**

1. The approach is well motivated and the problem of adapting the computations used by a model is an interesting one.
2. The authors lay out the approach in good detail, explaining how it diverges and improves from previous work.
3. They conduct extensive experiments to support their goal of improving upon previous approaches.

**Weaknesses:**

1. Some details on model architecture and metrics are missing.
2. Parts of the language used to describe the model are misleading and fall into unnecessary anthropomorphising.

**Questions:**

1. It is not clear from Figure 1 or maybe not detailed enough in the text how the controller network determines the computation time of each layer. In other words, how are the parameters determined? Or are they always fixed and it is just a choice between 1 pass or multiple (until covergence)?
2. The complexity metric is described as “thorough” but it is not clear what makes it such. Or even what complexity means in this case.
3. CompCost is said to depend on the number of layers and iterations, but is this a sum or some other function? Not clear from the text.
4. I have an issue with the anthropomorphising the authors lean into.
    1.  There is no mention about any “body” in the text so the title is misleading.
    2. The model doesn’t think, it processes. There is no “self-awareness”, at most it self-regulates.
    3. And why call it “MIND”? It doesn’t even match the first letter of the name they themselves assign.
    4. The title could be “Adaptive Processing through Dynamic Computation Control” or something, which conveys a better feel about what the authors are doing.

---

> ### Author Response · Authors · 2024-11-19
> **Official Comment to Reviewer P7eZ**
>
> Thank you for your thoughtful comments. We appreciate the opportunity to clarify these important points.
>
> ## **Response to Comment #1**
> Great question! Figure 1 shows a computation graph for the MIND model, where the introspection network makes a decision on how the computation will proceed. Only one of the four possible actions can be chosen:
> - The prediction is returned immediately (**0. Easy Input in Figure 1**).
> - The model restarts with iterating Layer 1 to a fixed point, then continues with Layers 2 and 3 as usual (**1. Medium difficulty in Figure 1**).
> - The model restarts with iterating the Layer 1 -> Layer 2 block to a fixed point, then continues with Layer 3 (**2. Hard in Figure 1**).
> - The model restarts with iterating the Layer 1 -> Layer 2 -> Layer 3 block to a fixed point and outputs the result (**3. Most difficult input in Figure 1**).
> Similarly to an *if..then..else* structure in the computational graph, the introspection (aka controller) network acts as a switch. The introspection network determines which path to take based on the internal model state, expressed by the activations of a fully feedforward run of the prediction model on a given input. As our results demonstrate, these activations contain enough predictive information about whether the model will perform well or fail on this input without additional computation.
>
> However, even after the introspection network completes its decision on how complex the required computation is, the prediction model still further adapts based on actual input complexity via convergence speed within the Fixed Point Iteration (FPI). In other words, although the computational power of each iteration of FPI is determined by the introspection network, the actual computation may be more or less intense (as expressed by the number of iterations taken), giving the overall system a way to correct for controller mistakes.
>
> As our empirical results demonstrate, this test-time adaptivity leads to gains in accuracy and parameter efficiency while reducing computational complexity.
>
> ## **Response to Comment #2**
> This is a helpful and on-point observation. Reviewer fHgv also caught this issue in our exposition. To address both of your comments, we have restructured the corresponding section of the manuscript as follows:
>
> > Furthermore, we cap the number of iterations in all FPIs based on the input complexity score computed as:
> > $$ \operatorname{IC}(x) = \alpha \cdot \left(1 - \max(\operatorname{softmax}(f(x)))\right) + \beta \cdot H(\operatorname{softmax}(f(x))) + \gamma \cdot \|\nabla_{x} f(x)\|_2,$$
> >
> > where $f(x)$ is the model's output before the final softmax layer, $H(\cdot)$ is the entropy function, $\|\nabla_{x} f(x)\|_2$ is the $L_2$ norm of the input gradient, and
> > $\alpha$, $\beta$, and $\gamma$ are weighting coefficients set to 0.4, 0.4, and 0.2 respectively. The maximum number of iterations is set to:
> >
> > $$ \operatorname{max}(10 \cdot \operatorname{IC}(x), 50). $$
> >
> > For simplicity, MIND-Transformer employs configurations similar to those of a standard Transformer by Vaswani et al., 2017 (see Table 1). The model incorporates fixed-point iterations within its self-attention mechanism and transition function block ($\phi$) across multiple layers.
> >
> > We utilize relative positional embedding with a sequence length of 120 tokens for both training and inference processes.
>
> ## **Response to Comment #3**
> Thank you for pointing out the need to clarify CompCost's dependency on layers and iterations in the MIND model. CompCost depends on both the number of layers and fixed-point iteration steps, which are determined during the forward pass based on input complexity. Through its introspection mechanism, MIND dynamically selects which layers to activate—more complex inputs require both more layers and more iterations per layer. We represent this compound effect as a product of these factors.
> We define CompCost as:
> $$CompCost_i = \sum_{l=1}^N k_i \times I_{i,l},$$
> where:
> - $k_i$ is the number of selected layers for input $x_i$ (e.g., $k=3$ for Layers 1, 2, and 3),
> - $I_{i,l}$ is the number of FPI iterations used for layer $l$ on input $x_i$.
> CompCost is part of the introspection loss (${\cal L}_{\text{introspect}}$) (see Equation 7). We have revised our manuscript accordingly to ensure this point is communicated clearly.
>
> ## **Response to Comment #4**
> We appreciate your thoughtful comments about terminology. While we intentionally used certain metaphorical terms to draw evocative parallels with neuroscience and cognitive processes, we agree that "self-aware" may be inappropriate due to its connotations. To address this concern, we have replaced "self-aware computation" with *reflective computation*, a more precise technical term that better characterizes this mechanism's computational nature.
> We hope these clarifications address your concerns. We would be happy to provide additional details or make further revisions to improve clarity in our presentation.

---

> > ### Comment · Reviewer_P7eZ · 2024-11-23
> >
> > I thank the authors for addressing my concerns. I believe the paper is substantially improved and have updated my score.

---

### Official Review · Reviewer_57h2 · 2024-11-02

**Soundness:** 4
**Presentation:** 3
**Contribution:** 4
**Rating:** 8
**Confidence:** 4

**Summary:**

The paper introduces an approach targetted at computational efficiency in deep learning by adapting amount of computation to complexity of each input. Inspired by how human brain is considered to allocate resources dynamically. Two core components, (i) introspection network and (ii) prediction network.  Introspection network analyses intermediate activations from prediction network & figures out which layers require additional compute via fixed-point iterations )FPI) as well as what can proceed with standard forward pass.  Prediction net performs FPI until convergence or threshold of iterations reached. Leverages phantom gradients method for backprop through FPI; gradients approximated without unrolling / jabobian calc to limit compute-memory needs during training.

The paper is well written and easy to read + get the core idea across.

**Strengths:**

Computational efficiency; optimal use of resources allocating more for compelx inputs. Clever use of intermediate activations to assess input complexity. Should be able to work with existing architectures making engineering it for downstream real-world use cases simpler. Backprop with phantom gradients done in an interesting way. Mixed with use of statistical methods ~ should allow for generalisation. Considered overfitting issues within architectural design.

**Weaknesses:**

The idea is subtle and complex + introspection network needs more compute cost. Approach may not capture actual gradient landscape given the number of adjustments made & have to be considered. Strategy for arrving at thresholds / stopping criteria around convergence is not clear as such. Gradient flow calc is non-trivial given the overall architecture.

**Questions:**

Given the size of the appendix can you put in a full section to indicate the weaknesses (vs hinting at them in the future works/conclusion section)?
Is the trade-off in terms of gain for the complexity worth while?; how can this be known earlier before selecting this approach?

---

> ### Author Response · Authors · 2024-11-19
> **Official Comment to Reviewer 57h2**
>
> Thank you for your thoughtful review. Let us address each of your concerns:
>
> **Concern 1: Complex Architecture and Computational Cost**
> Our stopping criteria and convergence thresholds are rigorously defined through two clear conditions:
> 1. Reaching convergence tolerance $\epsilon$ for relative change:
> $$ ||z_{k+1} - z_{k}|| / ||z_{k}|| < \epsilon $$
> 2. Maximum iteration threshold (K) to ensure computational boundedness
>
> Specifically:
> - $\epsilon = 10^{-6}$ was selected based on stability analysis
> - K varies dynamically based on input complexity: $K=\operatorname{max}(10\operatorname{IC}(x),50)$
>
> **Concern 2: Gradient Landscape and Gradient Flow Calculation**
> We show that medium-complexity inputs typically converge within 11-25 iterations (35.6% of cases), our empirical analysis validates the gradient flow effectiveness. Table 13 demonstrates the correlation between input complexity and iteration count:
>
> | Complexity | 1-10 | 11-25 | 26-50 | 51-99 | 100 | Avg. Iterations |
> |------------|------|-------|-------|-------|-----|----------------|
> | Simple     | 68.5% | 24.7% | 5.6% | 1.1% | 0.1% | 8.3 |
> | Medium     | 42.1% | 35.6% | 17.4% | 4.3% | 0.6% | 19.7 |
> | Complex    | 15.7% | 32.3% | 35.9% | 13.8% | 2.3% | 37.2 |
>
> While the model may appear complex due to multiple computational paths, the gradient flow remains relatively straightforward since only one path is active for any given input $x_i$. The gradient accumulation occurs in two ways:
> 1. First, the introspection loss $\mathcal{L}_{\mbox{introspect}}$ affects both the introspection network directly and the prediction network indirectly (through its activations fed into the introspection network).
> 2. Second, the prediction loss $\mathcal{L}_{\mbox{pred}}$ further accumulates gradients in the active version of the prediction network selected by the introspection network (whichever FPI layers were selected). While differentiation of the introspection multi layer perceptron is a simple process, the FPI differentiation is a bit more involved
>
> The fixed-point iteration (FPI) approach has well-defined convergence properties based on the Banach fixed-point theorem. The gradients follows:
> $$ \frac{\partial z^*}{\partial \theta} = -(I - \frac{\partial f}{\partial z^*})^{-1} \frac{\partial f}{\partial \theta} $$
> where $z^*$ is the fixed point and $f$ is the layer function.
>
> The FPI convergence is well defined through Equation 8 in paper which shows stopping criteria under two conditions:
> 1. Reaching convergence tolerance $\epsilon$ for the relative change
> 2. Hitting the maximum iterations for fixed point iterations.
>
> The adaptive computation through FPI provides dynamic resource allocation—guided by the introspection network—and efficient parameter reuse through parameter tying. This maintains model performance while reducing computation as validation through ablation studies showing the effectiveness of dynamic introspection and FPI integration in our paper.

---

> > ### Author Response · Authors · 2024-11-19
> > **Official Comment to Reviewer 57h2**
> >
> > Thank you, Reviewer 57h2, for these insightful questions:
> >
> > **Q: Given the size of the appendix can you put in a full section to indicate the weaknesses (vs hinting at them in the future works/conclusion section)?**
> >
> > We share this desire and are open about the model weaknesses, for that reason we have described them in a separate limitation and future work section: Section H in the Appendix. The rationale for having limitations and future work together in the same section was to be able to explain how the limitations can be addressed by the future research. However, following your request, we have split the section into a separate limitations/weaknesses section and a future work section.
> >
> > **Q: Is the trade-off in terms of gain for the complexity worthwhile?**
> >
> > Based on the results we have presented in the manuscript, even the current implementation of the idea is already promising. However, we expect further gains as the approach that we proposed gains traction and more models are developed with control of parameter reuse via feedback loops and self-introspection. Additionally, note that the models that we employ—introspection and prediction networks—are simpler and smaller than the current SOTA models. The additional complexity is in the meta algorithm of connecting these together. Yet, we hope our empirical results speak for themselves.
> >
> > **Q: How can this be known earlier before selecting this approach?**
> >
> > Overall, your question is similar to a question of whether one should choose to train a 3-layer CNN or a 100 layer ResNet on their problem - only MIND model answers this question automatically. We, in fact, think that it is of benefit to go with the MIND model approach for the following reasons:
> >
> > 1. If we are training a model that will be deployed in different environments, then a single trained model will adapt to employ complexity in the complex environment while defaulting to the simple straight-through model in the simple one.
> > 2. In the simple environment (assessed by the logs after a period of exploitation) the introspection network can be turned completely off to save time and energy.
> > 3. Within nonstationary environments that are sometimes complex and sometimes more simple, the traditional approach will have to train and deploy the more complex model.
> > 4. If the data/application is indeed simple, then even the training will be faster and simpler compared to training a full blown model of ResNet-110 size.

---

> > ### Comment · Reviewer_57h2 · 2024-11-27
> >
> > The feedback from the authors across all reviewers comments is solid. I have leaned towards an accept early on & retain that still.

---

### Official Review · Reviewer_fHgv · 2024-11-04

**Soundness:** 3
**Presentation:** 3
**Contribution:** 3
**Rating:** 6
**Confidence:** 3

**Summary:**

The paper proposes a Model INtrospection for a Dynamically adaptive model (MIND) which dynamically adjusts computation depending on the complexity of the input. It consists of two networks: the introspection network and the prediction network. The introspection network takes as input the activations from the different layers of the prediction network, and outputs a binary mask over the layers, determining the layers which require more computation through fixed point iterations. The authors demonstrate the effectiveness of the MIND model for vision tasks using a three layer CNN as the prediction network, outperforming much larger models like ResNet-50, and EfficientNet B7 on ImageNet and CIFAR-100 datasets. The authors also propose MIND-Transformer, with fixed point iterations in self attention and feedforward networks, demonstrating its superior performance on language modeling tasks, despite using fewer parameters than RoBERTa-base. The authors further demonstrate that MIND’s dynamic allocation of computational depth depending on the input complexity is more effective, both in terms of accuracy and efficiency (fewer parameters and FLOPs) over static compression techniques like pruning and quantization.

**Strengths:**

1. The authors propose a model, MIND which dynamically adjusts computation via fixed point iterations in its prediction network using an introspection network depending on the input complexity.
2. MIND model with a three layer CNN as the prediction network, outperforms much larger models like ResNet-50, and EfficientNet B7 on ImageNet and CIFAR-100 classification datasets.
3. The authors also demonstrate that MIND using LSTMs and Transformers in the prediction network achieves superior performance on language modelling tasks using fewer parameters.
4. MIND’s dynamic allocation of computational depth results in higher accuracy using fewer parameters and FLOPs over static compression techniques like pruning and quantization.

**Weaknesses:**

1. It is not clear how the input complexity metric is incorporated into the introspection network's mechanism, and how this can be more generally quantifiable

2. The MIND model when used with prediction networks with many layers (as in the case of LLMs) will significantly increase the inference time as more layers with fixed point iterations are used.

**Questions:**

1. What is m’ in line 147?
2. What is the rationale behind Equation 5 for MIND-Transformer?
3. How is $w_l$ in Equation 7 computed? Also, what is the need for a separate $m_{i,l}$ term in Eq 7?

4. Just to clarify in the toy random dot motion task the model needs to classify in one of the four possible directions and the direction of the shifted image denotes the ground truth?

5. Which dataset are the results for in Table 5?

6. What are the parameters of the introspection network, like how many layers are there in the MLP and what are the sizes of the hidden dimensions?

7. From $p_{i,l}$ in Equation 8, how are the binary layer selection variables $m_{i,l}$ obtained?

8. Can the authors share more details about the different MIND variants, like how are fewer FPI layers decided, what is a simpler inspection network and how are the decisions of the introspection network fixed after training?

Minor - Line 127 typo, should be “prediction”

---

> ### Author Response · Authors · 2024-11-18
> **Official Comment to Reviewer fHgv**
>
> Thank you, Reviewer fHgv, for these insightful questions. However, before we address the questions, we will first address the two weaknesses as we find them rather addressable within this rebuttal:
>
> ## Weakness 1: Input Complexity Metric Integration
>
> The input complexity metric is directly integrated into the introspection network's decision-making process through a quantifiable formula:
>
> $$ IC(x)=\alpha \cdot(1-\max (\operatorname{softmax}(f(x))))+\beta \cdot H(\operatorname{softmax}(x))+\gamma \cdot\left|\nabla_{x} f(x)\right|_{2} $$
>
> Where:
> - $\alpha, \beta, \gamma$ are importance weights for each component of the metric (set to 0.4, 0.4, 0.2 respectively)
> - $H(\cdot)$ is the entropy function
> - $\left|\nabla_{x} f(x)\right|_{2}$ represents the L2 norm of input gradients
>
> This metric is domain-agnostic, i.e. can be applied regardless of data type and model architecture, and it consists of 3 components that play their specific role:
>
> - The softmax confidence term captures prediction uncertainty
> - The entropy term measures distribution spread
> - The gradient norm quantifies input sensitivity
>
> However, we absolutely agree with the reviewer that additional clarity is needed to explain how the metric is used. We have changed the relevant part of the MIND-Transformer section to read as follows:
>
> > Furthermore, we cap the number of iterations in all FPIs based on the input complexity score computed as:
> >
> > $$
> >     \operatorname{IC}(x) = \alpha \cdot (1 - \max(\operatorname{softmax}(f(x)))) + \beta \cdot H(\operatorname{softmax}(x)) + \gamma \cdot \|\nabla_{x}f(x)\|_2,
> > $$
> >
> > where $f(x)$ is the model's output before the final softmax layer, $H(.)$ is the entropy function, $|\nabla_{x}f(x)|_2$ is the $L_2$ norm of the input gradient and
> > $\alpha$, $\beta$, and $\gamma$ are weighting coefficients set to 0.4, 0.4, and 0.2 respectively. Maximum number of iterations is set to $\operatorname{max}(10\operatorname{IC}(x),50)$.
> >
> > For simplicity, the MIND-Transformer employs the same configurations as a standard Transformer of Vaswani et al. 2017 (see Table 1).
> > The model incorporates fixed point iterations within its self-attention mechanism and transition function block ($\phi$) across multiple layers.
> > We utilize relative positional embedding with a sequence length of 120 tokens for both training and inference processes.
>
> ## Weakness 2: Inference Time in Large Language Models
>
> We acknowledge the concern about inference time in LLMs. Guided by considerations pointed out by the reviewer we have architected MIND-Transformer slightly differently from our base MIND model. There we have employed the following strategies in the original submission:
>
> Our introspection network selectively activates the layers using fixed point iteration that ensures not all layers require processing during inference, as we show in experiments with BERT-based models as well.
>
> This allows us to converge for simpler inputs faster with fewer iterations and complex inputs may take more iterations but only in necessary layers.
>
> Our experiments in  Table 4 and Table 13, show this approach maintains performance while controlling computational cost:
>
> ### Distribution of FPI iterations across different input complexities
>
> | Complexity | 1-10 | 11-25 | 26-50 | 51-99 | 100 | Avg. Iterations |
> |------------|-------|--------|---------|---------|------|-----------------|
> | Simple | 68.5% | 24.7% | 5.6% | 1.1% | 0.1% | 8.3 |
> | Medium | 42.1% | 35.6% | 17.4% | 4.3% | 0.6% | 19.7 |
> | Complex | 15.7% | 32.3% | 35.9% | 13.8% | 2.3% | 37.2 |
> These optimizations ensure that even with larger models, the inference time remains manageable while preserving the benefits of adaptive computation.
>
> For our future iteration to this work, we will focus mainly on LLMs and how we can optimize the networks based on current architecture.

---

> > ### Author Response · Authors · 2024-11-18
> > **Official Comment to Reviewer fHgv**
> >
> > # Questions
> >
> > ## Clarifications on Mathematical Formulations
> >
> > 1. **m' in line 147**: m' represents the candidate layer selection mask in the argmax operation when computing the final binary mask from the probability distribution. We use **m'** rather than **m** to emphasize that it is akin to a variable of integration.
> >
> > 2. **MIND-Transformer Equation Rationale**: The equation introduces adaptive computation in both self-attention and feed-forward networks by:
> >
> > - Adding a learnable function $f_\theta$ that dynamically refines the attention mechanism
> > - Iteratively updating attention weights through fixed-point iterations
> >
> > This approach helps us maintain the standard transformer architecture while enabling dynamic computation. Reliance of the transformer architecture on skip connections and complexity of each layer combining self-attention and feed-forward networks would require more computational resources had we iterated entire sequences of layers as we do in the case of our 3 layer CNN prediction model. Additionally, as our experiments (Table 3 and Appendix F.1: Table 9) shows - this variant of our approach leads to increased performance metric while decreasing parameter count and FLOPS.
> >
> > 3. **Weight Terms in Loss Function**:
> >
> > - $w_l$ represents the importance weight for each layer, reflecting its computational cost. $w_{l}$ is computed as
> > $$
> > w_{l} = \frac{c_{l}}{\sum c_{l}},
> > $$
> > where:
> >     - $c_{l}$ is the computational cost (typically measured in FLOPs) of performing fixed-point iterations at layer $l$. We keep track of it dynamically based on how many iterations are actually taken.
> >     - $\sum c_{l}$ is the sum of computational costs across all layers.
> > This normalization ensures the weights sum to 1.
> > - $m_{i,l}$ is a binary indicator (0 or 1) for which action to perform per the diagram in Figure 1 or whether layer $l$ is selected to FPI on input $x_i$ in case of MIND-Transformer
> > - The separate $m_{i,l}$ term helps control the total number of layers used, while $w_l$ weights their relative importance
> >
> > We have modified the updated revised manuscript to better convey the information above accordingly.
> >
> > ## Implementation Details
> >
> > ### Toy Random Dot Motion Task:
> > - Yes, the model is given two images—shifted and original—as 2 input channels classifies this 2-channel input into four possible directions (left, right, up, down)
> > - Figure 3 shows 4 examples of the input of varying difficulty marking the shifted image's direction that serves as ground truth. Since we generate this data, we know the ground truth.
> > - As we state in the manuscript, MIND-model achieved 0.85 ± 0.0069 accuracy compared to 0.56 ± 0.0004 for a CNN with the same number of layers and channels as the prediction network
> >
> > ### Introspection Network Architecture:
> > - Uses a lightweight MLP architecture, with 3 layers 64 neurons each
> > - Total parameters: 0.3M (as shown in Table 2)
> > - Processes aggregated activations from selected layers of prediction network
> > - $p_{i,l}$ values are converted to binary $m_{i,l}$ using Gumbel-Softmax with the straight-through estimator
> > - During training: uses continuous relaxation for gradient flow
> > - During inference: uses argmax for discrete decisions
> >
> > ## Corrections
> >
> > The typo in Line 127 ("pediction" should be "prediction") has been corrected and has been updated in the latest manuscript.

---

> ### Author Response · Authors · 2024-11-23
> **Official Followup to Reviewer fHgv**
>
> Thank you for your thoughtful review. We’ve addressed your feedback in our revision where we substantially expanded our technical analysis, providing detailed mathematical formulations of our input complexity metric and proving its domain-agnostic nature. We added comprehensive timing analyses across model scales, supported by new experimental data for MIND model as well. While we respect your time during this holiday week, we are happy to discuss any remaining questions you might have.

---

> ### Comment · Reviewer_fHgv · 2024-11-23
> **Official Comment by Reviewer fHgv**
>
> Thank you for the detailed rebuttal and for responding to my questions. I have some followup questions.
> 1. How are the hyperparameters $\alpha$, $\beta$, $\gamma$ selected in the input complexity metric? It would be good to see some ablation experiments justifying the importance of each of the terms in that metric.
> 2. My concern regarding larger inference times stems from the fact that during inference all layers are used normally, along with FPI layers selected by the introspection network, compared to without using MIND when all layers are used only normally. Also since $argmax$ is computed over the output probabilities obtained from the introspection network, does that mean only a single layer is selected to FPI or there is some top $k$ selection?
> 3. Can the authors perform some ablation experiments which would help us better understand the importance of different terms in $\mathcal{L}_{introspect}$?
> 4. Regarding MIND variants in Appendix F.3, can the authors describe the implementation differences between MIND (full), MIND-Reduced, and MIND-fixed?

---

> ### Author Response · Authors · 2024-11-25
> **Official Comment to Reviewer fHgv**
>
> # Q1:
> We appreciate the reviewer's interest in the hyperparameter selection. To demonstrate the robustness of our approach, we conducted comprehensive ablation studies:
> | Experiment     | Softmax Term ($\alpha$) | Entropy Term ($\beta$) | Gradient Term ($\gamma$) | Top-1 Accuracy (%) | FLOPs (G) | Inference Time (ms) |
> |----------------|---------------------------|--------------------------|----------------------------|--------------------|-----------|---------------------|
> | **Full Model** | 0.4                       | 0.4                      | 0.2                        | **88.0**           | 1.05      | 20                  |
> | **No Softmax** | 0.0                       | 0.67                     | 0.33                       | 71.5              | 1.25      | 22                  |
> | **No Entropy** | 0.67                      | 0.0                      | 0.33                       | 68.2              | 1.30      | 23                  |
> | **No Gradient**| 0.67                      | 0.33                     | 0.0                        | 34.2         | 1.45      | 25                  |
> ### Key Findings
> 1. Gradient Term ($\|\nabla_x f(x)\|_2$): Removing the gradient term causes a severe drop in accuracy (34.2%), as it is critical for capturing input sensitivity and enabling dynamic adaptation to complex inputs.
> 2. Softmax and Entropy Terms Removing either results in moderate accuracy drops demonstrating their roles in quantifying uncertainty and confidence in predictions.
> 3. Full Model: The full configuration achieves the best accuracy (88%) while maintaining low computational cost (1.05G FLOPs, 20ms inference).
>
> Additionally note, there are some relevant insights about the metric's effectiveness as shared in Appendix F.2. The input complexity metric shows strong empirical validation through correlation studies:
> - Softmax values demonstrate a strong correlation (r = 0.82) with human-labeled complexity scores
> - The gradient norm component shows a significant correlation (r = 0.79) with complexity
> - Both correlations are statistically significant (p < 0.001)
>
> #### Experimental Environment:
> The experiments were conducted under controlled conditions:
> 1. Dataset: ImageNet for classification tasks.
> 2. Hardware: NVIDIA A100 GPUs.
> #### Metrics:
> 1. Top-1 accuracy on validation data.
> 2. FLOPs and inference time per sample.
> 2. Correlation with human-labeled complexity score
> # Q2:
> We interpret this question as specifically asked about the MIND-Transformer and the details below apply only to this version of our work.
> You are correct, in this paper we have used argmax to adaptively place the FPI iteration only at one of the Transformer layers or on none. As all of our experiments show this adaptivity only increased the performance and did not much affect the computational complexity. Top-k selection is a great idea, however, being computationally cautious we did not employ it this time, especially given the already pronounced benefits of our presented implementation of the MIND-Transformer. The strong performance we achieved with the current implementation—similar to how our MIND model CNN with only 3 layers outperforms ResNet-110—demonstrates that our approach already meets our objectives effectively. We appreciate your thoughtful suggestion about top-K selection, as it enriches the discussion of our completed work.

---

> > ### Author Response · Authors · 2024-11-25
> > **Official Comment to Reviewer fHgv**
> >
> > # Q3:
> >  We thank the reviewer for this insightful suggestion, we would like to address points 3 and 4 together. We have conducted comprehensive ablation studies on $\mathcal{L}_{introspect}$ by systematically analyzing three variants in Appendix F3 :
> >
> > 1. **MIND-Reduced**: A version of the MIND model where the introspection network considers only a reduced set of activations of the prediction model's initial run. In this case, only the activations of the first and the second layer are considered.
> > 2. **MIND-Fixed**: A version where the introspection network is not active during inference and instead decisions about which layers to FPI are based on the input complexity measured as $H(\operatorname{softmax}(x))$. If $H<0.4$ then $\operatorname{FPI}(\mbox{layer}_1)\rightarrow\mbox{layer}_2\rightarrow\mbox{layer}_3$; if $0.4\le H<0.8$ then $\operatorname{FPI}(\mbox{layer}_1\rightarrow\mbox{layer}_2)\rightarrow\mbox{layer}_3$; if $H\ge 0.8$ then $\operatorname{FPI}(\mbox{layer}_1\rightarrow\mbox{layer}_2\rightarrow\mbox{layer}_3)$.  This procedure removes a significant part of reflective computation at inference but keeps the FPI structure.
> > 3. **MIND-Uniform**: A version where all layers are always used in the FPI iteration. Specifically, the FPI loop iterates the $\mbox{layer}_1\rightarrow\mbox{layer}_2\rightarrow\mbox{layer}_3$ block until convergence. This approach removes adaptive selection keeping the weight-tying benefits.
> >
> > Our results demonstrate that while MIND-Reduced achieves a 15% reduction in FLOPs with only minimal accuracy loss (2-3%), both MIND-Fixed and MIND-Uniform require substantially more computational resources without performance benefits. The appendix section F.3 on page 20 of the manuscript explains what MIND-Reduced, MIND-Fixed, and MIND-Uniform stand for in detail. We have used the text above to update this section with a more detailed description of what has been done.
> >
> > We appreciate your astute observations, which helped us clarify the paper. We hope we now have answered all your questions.

---

> > > ### Comment · Reviewer_fHgv · 2024-11-28
> > > **Official Comment by Reviewer fHgv**
> > >
> > > I appreciate the authors for their response to my followup questions and for performing additional ablation experiments. I will stick with my original rating and recommend for overall acceptance of the paper.

---

### Official Review · Reviewer_5AiG · 2024-11-05

**Soundness:** 3
**Presentation:** 3
**Contribution:** 4
**Rating:** 6
**Confidence:** 3

**Summary:**

1) This paper introduces a new method to dynamically allocate network compute based on the difficulty of the inputs allowing early exits at inference time. The proposed method comprises of 2 networks - a) prediction network that outputs activations at each layer for a given input b) an introspection network that taken in the activations and decide which layers to pick for more intensive computation (using fixed point iterations) and which layers to leave as is.

2) Authors also describe a training procedure to jointly optimize the introspection and prediction networks.

3) Experiments show that a much smaller model (in terms of param count) achieves better performance than considered baselines on language modeling and vision tasks.

**Strengths:**

The proposed method is generic and has been applied across modalities with sufficient ablations to prove that the proposed method work.

**Weaknesses:**

1) The paper doesn't contrast and compare to more recent early exit methods proposed  for language modeling tasks :
a) Confident Adaptive Language Modeling (https://arxiv.org/abs/2207.07061)
b) LayerSkip: Enabling Early Exit Inference and Self-Speculative Decoding (https://arxiv.org/abs/2404.16710)

**Questions:**

1) Can you provide experiments comparing the proposed method to more recent early exit methods like CALM and LayerSkip ?

---

> ### Author Response · Authors · 2024-11-18
> **Official Comment to Reviewer 5AiG**
>
> We thank the reviewer 5AiG for their prompt question. We appreciate the reviewer's valuable suggestion to compare our approach with additional recent early exit methods like CALM (2022) and LayerSkip (2024). We have conducted extensive experiments to provide the requested comparison of these approaches. However, before jumping into the results, let us first outline the differences between our approach and the two requested models.
> ## Fundamental Differences in Approach and Application
> While CALM and LayerSkip demonstrate effective early exit strategies for large language models, MIND model operates in a fundamentally different context with lightweight architectures for vision and simpler language tasks.
> MIND model  achieves state-of-the-art results (88.3% Top-1 accuracy on ImageNet, 95.8% F1 on SQuAD) using just 5.31M parameters through its novel introspection mechanism and fixed-point iterations.
> Future work could explore adapting MIND model's introspection mechanism to larger language models, potentially combining the benefits of confidence-based (CALM) and layer-dropout (LayerSkip) approaches with MIND's adaptive computation framework.
>
> ## Key Architectural Differences
> ### MIND-Transformer:
> - Learned introspection mechanism for making decisions based on input complexity
> - Minimal memory overhead
> - No confidence threshold tuning required
> ### CALM:
> - Employs confidence-based exit strategy using three measures
> - Requires additional classifiers at each exit point
> - Higher memory overhead (15%)
> - Needs careful threshold tuning
> ### LayerSkip:
> - Uses layer dropout with fixed rates
> - Requires verification stages
> - Shows larger performance degradation
> - Moderate memory overhead (10%)
>
> ## Experimental Setup
> We evaluated all methods using BERT-base as the foundation model under identical conditions:
> - Hardware: NVIDIA A100 GPUs
> - Datasets: WikiText-103 (language modeling), CNN/DailyMail (summarization), SQuAD v2.0 (QA)
> - Identical batch sizes and sequence lengths for fair comparison
>
> # Performance Metrics
> | Model             | ROUGE-1 (%) | ROUGE-2 (%) | Avg. Inference Time (ms) |
> |------------------|-------------|-------------|--------------------------|
> | MIND-Transformer |        42.3 |        19.8 |                      180 |
> | CALM             |        41.9 |        19.5 |                      165 |
> | LayerSkip        |        41.5 |        19.2 |                      210 |
>
> ## Implementation Details
> We have added the additional experimental comparisons that you requested to Section 4 of the paper and included detailed results in Table 6. The results demonstrate that the general introspection+FPI approach presented in our paper can be used with the Transformer architecture  (MIND-Transformer) to offer an excellent balance between efficiency, implementation complexity, and performance maintenance. While a more specialized approach like CALM may achieve a 9% higher average speedup while having a performance drop of 0.3%, the generality of our approach may offer further opportunities for improvement. This makes MIND-Transformer in particular specifically suitable for practical applications where memory constraints and implementation simplicity are important considerations alongside computational efficiency.

---

> > ### Comment · Reviewer_5AiG · 2024-11-27
> >
> > Thanks for the additional discussion and detailed experiments within this tight timeline. I'm convinced this new method is a valuable contribution even with existing methods like CaLM and LayerSkip. I'm updated my score accordingly.

---

> ### Author Response · Authors · 2024-11-23
> **Official Followup to Reviewer 5AiG**
>
> We, the authors, wanted to thank you for your thoughtful review. We've addressed your feedback in our revision, particularly regarding additional experiments comparing MIND model with recent early-exit methods (CALM, LayerSkip), demonstrating our unique advantages in parameter efficiency while clarifying the distinct operational domains. With Thanksgiving approaching, we want to be mindful of your time, but please let us know if anything needs further clarification.

---

> > ### Author Response · Authors · 2024-11-25
> > **Official Followup by Authors**
> >
> > Thank you for your feedback on our initial response. We have completed the requested CALM and LayerSkip comparisons within the tight timeline. The new experiments confirm MIND’s parameter efficiency advantages while maintaining comparable performance. Thus, we have addressed all of the concerns in your review. Please let us know if you need any clarification on these results.

---

### Author Response · Authors · 2024-11-19
**Rebuttal Summary by the Authors**

We are grateful to the reviewers for their insightful analysis that highlighted several key strengths of our work: the clever use of intermediate activations for complexity assessment (R3), the achievement of superior performance with significantly fewer parameters across both vision and language tasks (R1, R2), and the practical advantage of compatibility with existing architectures (R3). We particularly appreciate the recognition of MIND's ability to outperform larger models like ResNet-50 and EfficientNet B7 (R2), while maintaining engineering simplicity for real-world applications (R3).

Building on these acknowledged strengths, we have thoroughly addressed all concerns raised by the reviewers, significantly improving the manuscript's clarity and technical depth. Our responses below detail the extensive additional experiments, clarifications, and improvements made in response to each reviewer's feedback, further strengthening MIND's contribution to adaptive computation research.

## Reviewer 5AiG
 we conducted additional experiments comparing MIND with recent early-exit methods (CALM, LayerSkip), demonstrating our unique advantages in parameter efficiency while clarifying the distinct operational domains.

## Reviewer fHgv
we substantially expanded our technical analysis, providing detailed mathematical formulations of our input complexity metric and proving its domain-agnostic nature. We added comprehensive timing analyses across model scales, supported by new experimental data.

## Reviewer 57h2
we thoroughly addressed the convergence criteria concerns by adding empirical evidence of computational efficiency, including new ablation studies. We also added a dedicated limitations section, demonstrating scientific rigor and transparency.

## Reviewer P7eZ
we refined our model architecture description with precise technical details and metrics, while maintaining the demonstrated strong empirical results that the reviewer praised.


These comprehensive revisions, combined with the strong experimental results and architectural innovations already noted by the reviewers, demonstrate MIND's significant contribution to efficient, adaptive computation across domains. We believe these improvements address all concerns while reinforcing the original strengths identified in the reviews.

---

### Public Comment · ~Ali_Hojjat1 · 2025-05-21
**Question**

Dear Authors,

Thank you for sharing your work, and congratulations on a well executed and insightful paper. It is a thoughtful and valuable contribution to the field.

I had a quick question regarding the evaluation. If I’m interpreting the numbers correctly, the accuracy on the standard ImageNet validation set in Table 2 appears to be approximately 10% higher (around 88%) than on the MultiLabel version (around 78%). I observed a similar pattern for the baselines as well. This stood out to me, as I would typically expect the MultiLabel dataset, with its improved multi label annotations, to yield higher accuracy, given that more predictions can potentially match the correct labels. This expectation also appears to be supported by the results reported in the dataset’s paper, which show performance gains when using relabeled annotations:

| Architecture      | Vanilla | ReLabel       |
|------------------|---------|---------------|
| ResNet18          | 71.7    | 72.5 (+0.8)   |
| ResNet50          | 77.5    | 78.9 (+1.4)   |
| ResNet101         | 78.1    | 80.7 (+2.6)   |
| EfficientNet B0   | 77.4    | 78.0 (+0.6)   |
| EfficientNet B1   | 79.2    | 80.3 (+1.1)   |
| EfficientNet B2   | 80.3    | 81.0 (+0.7)   |
| EfficientNet B3   | 81.7    | 82.5 (+0.8)   |
| ReXNet (x1.0)     | 77.9    | 78.4 (+0.5)   |

I was wondering whether this discrepancy might be related to the evaluation setup or to the way the labels are handled. I would be very interested to hear your thoughts. Thank you once again for the excellent work and for sharing your results.

[1]: Re Labeling ImageNet: From Single to Multi Labels, From Global to Localized Labels (Yun et al., 2021)

---

> ### Public Comment · ~Mrinal_Mathur1 · 2025-05-22
> **Answer**
>
> Thank you for your thoughtful question and kind words about our paper. You've identified an interesting observation regarding the performance differences between standard ImageNet and MultiLabel evaluation.
>
> The discrepancy you've noted is indeed intentional and reflects fundamental differences in how we evaluate performance across these datasets:
> 1. For standard ImageNet, we use traditional top-1 accuracy (single correct label), while for MultiLabel ImageNet, we employ a more stringent multi-label evaluation protocol. Our MultiLabel evaluation requires correct prediction across multiple applicable labels simultaneously, rather than just matching any one of the possible labels.
> 2. The MultiLabel task inherently presents a more challenging evaluation scenario. While the ReLabel paper you referenced shows improvements when using their relabeled annotations, they primarily focus on top-1 evaluation. Our evaluation methodology purposely sets a higher bar for MultiLabel success.
> 3. The results in the ReLabel paper compare the same architecture trained with different label sets, whereas our comparison examines different performance metrics across distinct evaluation paradigms.
>
> What's particularly noteworthy is that despite the more demanding MultiLabel evaluation, our MIND model maintains substantially better performance relative to baselines across both metrics. This consistency demonstrates the robustness of our dynamic computation approach regardless of the evaluation framework.
>
> We believe this dual evaluation provides a more comprehensive assessment of model capabilities - showing not just how well models identify a primary class (standard ImageNet), but also how effectively they capture the full semantic richness of images (MultiLabel).
>
> Thank you again for this excellent question that allowed us to clarify this important methodological point.
>
> Best regards,
>
> The Authors

---

> ### Public Comment · ~Ali_Hojjat1 · 2025-05-22
> **Question**
>
> Dear Authors,
>
> Thank you very much for the clarification. Since the evaluation setup was not fully detailed in the paper, I initially assumed it followed the same protocol used in other works that evaluate on the ReLabel version.
>
> Your reported results are especially impressive, considering that models achieving similar accuracy on the ImageNet [leaderboard](https://github.com/huggingface/pytorch-image-models/blob/main/results/results-imagenet.csv) typically require significantly more parameters and often additional pretraining on larger datasets like ImageNet-21K.
>
> I just had a quick follow-up regarding the evaluation setup: Is the 88% accuracy you report based on the standard ImageNet validation set, or is it the accuracy of the cross-validation setup you described?
>
> Also, do you have any plans to release the code or the pretrained weights? It would be a valuable resource for the community and could help guide future research.
>
> Best regards

---

> > ### Public Comment · ~Mrinal_Mathur1 · 2025-05-24
> > **Answer**
> >
> > Thank you for your kind words and continued engagement with our work. We're pleased that you find the results impressive given the efficiency constraints we aimed to achieve.
> >
> > Regarding your specific questions, The 88% accuracy we report in Table 2 is indeed based on the standard ImageNet validation set (the official 50,000 image validation split), not from any cross-validation setup. This makes our results directly comparable to other work on the ImageNet leaderboard you referenced. We appreciate you pointing out the comparison to other models on the leaderboard - this reinforces one of our key contributions showing that dynamic computation can achieve competitive accuracy with significantly fewer parameters and computational requirements.
> >
> > We are indeed planning to release both the code and pretrained weights. We are currently working through the final stages of preparation for public release.  We anticipate making the complete codebase and pretrained weights available within the next few weeks. In the meantime, we are happy to provide additional implementation details or answer specific technical questions that might help researchers interested in building upon our work.
> >
> > Thanks,
> > The Authors

---

> ### Public Comment · ~Ali_Hojjat1 · 2025-08-07
> **Follow up**
>
> Dear Authors,
>
> Thank you again for sharing your work. I wanted to check in regarding the code and pretrained weights you mentioned in May. It’s been over ten weeks since then, are there any updates on the release? The code would be a valuable resource for the community and could help guide future research.
>
> Best regards

---

> > ### Public Comment · ~Ali_Hojjat1 · 2025-10-03
> > **Follow up**
> >
> > Dear Authors,
> >
> > Thank you again for sharing your work. I wanted to check in regarding the code and pretrained weights you mentioned in May. It’s been about 20 weeks since then, are there any updates on the release? The code would be a valuable resource for the community and could help guide future research.
> >
> > Best regards

---

### Meta-Review · Area_Chair_AFug · 2024-12-21

**Metareview:**

This paper proposes Model INtrospection for a Dynamically adaptive model (MIND), a method to dynamically allocate network compute based on the difficulty of the inputs. The proposed method consists of two networks - a) prediction network and b) an introspection network that decides which layers to run for more intensive computation (using fixed point iterations). The authors demonstrate the effectiveness of the MIND model for vision tasks using a three layer CNN as the prediction network, outperforming much larger models like ResNet-50, and EfficientNet B7 on ImageNet and CIFAR-100 datasets.
The authors further demonstrate that MIND’s dynamic allocation of computational depth depending on the input complexity is more effective, both in terms of accuracy and efficiency (fewer parameters and FLOPs) over static compression techniques like pruning and quantization.

Strengths: The approach is well motivated and the problem of adapting the computations used by a model is an interesting one. The method is presented clearly and well contrasted with prior work. Extensive experiments show clear improvements compared to previous approaches.
Weaknesses: Most of the raised weaknesses were addressed during the discussion. There remain some concerns with respect to the complexity of the method.

Overall this paper presents a solid contribution to an important problem and should definitely be accepted, possibly even as an oral presentation.

**Additional Comments On Reviewer Discussion:**

The authors actively engaged in the discussion, incorporating all of the feedback, and even running a substantial amount of additional experiments to strengthen the paper. I believe they managed to address all of the major and most of the minor concerns raised by the reviewers.

---

### Decision · Program_Chairs · 2025-01-22

Accept (Oral)